# Bayesian Neural Networks with Domain Knowledge Priors

## Abstract

Bayesian neural networks (BNNs) have recently gained popularity due to their ability to quantify model uncertainty in prediction. However, specifying a prior for BNNs that accurately captures relevant domain knowledge is often extremely challenging. In this work, we propose a framework for integrating general forms of domain knowledge (i.e., any knowledge that can be represented by a loss function) into a BNN prior through variational inference, while enabling computationally efficient posterior inference and sampling. Specifically, our approach results in a prior over neural network weights that assigns high probability mass to models that better align with our domain knowledge, leading to posterior samples that also exhibit this behavior. In a semi-supervised learning setting, we show that BNNs using our proposed domain knowledge priors outperform those with standard priors (e.g., isotropic Gaussian, Gaussian process), successfully incorporating diverse types of prior information such as fairness, physics rules, and healthcare knowledge and achieving better predictive performance. We also present techniques for transferring the learned priors across different model architectures, demonstrating their broad utility across many tasks.

## 1 Introduction

While recent advances in deep learning have led to strong empirical performance in many real-world settings, it is crucial for deep learning models to faithfully represent the uncertainty in their predictions and avoid making incorrect predictions with high confidence, especially in safety-critical domains (e.g., healthcare, criminal justice). Unfortunately, prior works show that deep learning models trained via empirical risk minimization often make errors with high confidence at test time, especially on data points that differ from those observed in the training data distribution (Hendrycks & Gimpel, 2016; Hendrycks et al., 2021). Moreover, these models often inherit undesirable biases present in their training data (Larson et al., 2016; Obermeyer et al., 2019), motivating the development of an approach for incorporating prior knowledge into model training to mitigate such issues.

A principled approach to achieving both good predictive performance and a faithful representation of predictive uncertainty is to use Bayesian neural networks (BNNs; MacKay, 1992; 1995; Neal, 1996; Wilson & Izmailov, 2020; Papamarkou et al., 2024). In the Bayesian setting, selecting a good prior is crucial, and its misspecification for BNNs can force the posterior distribution to contract to suboptimal regions of the weight space (Grünwald & van Ommen, 2017; Gelman et al., 2017; 2020; Fortuin, 2022), resulting in suboptimal posterior predictive performance. Ideally, the prior should well-reflect what relevant domain knowledge (e.g., physics rules) specifies as plausible functions for a given prediction problem and help mitigate any undesirable biases learned from the training data.

However, the high-dimensionality of the weight space and the nontrivial connection between the weight and function spaces make specifying a prior that reflects domain knowledge challenging (Nalisnick, 2018; Fortuin, 2022). Due to such difficulties, uninformative priors that enable tractable sampling and approximate inference are typically used in practice. The most widely used uninformative weight-space prior is the isotropic Gaussian prior (Hernández-Lobato & Adams, 2015). Recent works propose to directly specify a function-space prior (e.g., via Gaussian processes (GPs)) to encode functional properties such as smoothness and periodicity (Sun et al., 2019). However, existing forms of informative priors are not flexible enough to represent broader forms of domain knowledge.

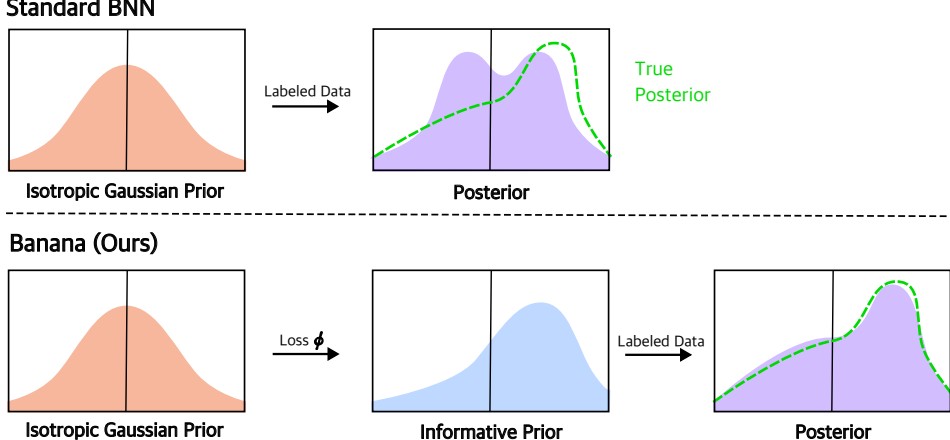

Figure 1: Our framework (Banana; bottom) compared to standard practice (top) for training BNNs. We propose a variational inference approach that learns an informative prior by updating the isotropic Gaussian prior with relevant domain knowledge via a loss function $\phi$ (Section 4). Our informative prior learned using unlabeled data helps encourage models that exhibit desirable behavior (Section 5).

In this work, we propose a novel approach for incorporating much more *general* forms of domain knowledge into BNN priors in a semi-supervised learning setting. The key challenge lies in both how to formulate and incorporate such knowledge into the prior and to ensure that this prior enables to computationally tractable posterior inference and sampling. In particular, we focus on domain knowledge for which we can formulate a *loss function* $\phi$, such that it captures how well a particular model aligns with the given knowledge. We show that various forms of domain knowledge can be represented in this form (Section 5.1). For example, for a physics rule, we can define the loss function to measure how much a model's prediction of the state of a physical system violates the law of conservation of energy. As another example, if we want a vision model to ignore the background of an image, we can define the loss to be the norm of the gradient of the model's prediction with respect to the background pixels of the image. To obtain an informative prior that incorporates such domain knowledge, we propose a variational inference approach to learn a low-rank Gaussian distribution that puts higher probability mass on model weights with low values of the loss $\phi$ on unlabeled data. The low-rank Gaussian structure of the informative prior enables computationally efficient posterior inference. We emphasize that with existing approaches for specifying informative priors, it is not clear how to incorporate similar forms of prior knowledge.

We demonstrate that using our learned informative priors for posterior inference in BNNs not only ensures better alignment with domain knowledge (i.e., lower values of $\phi$) but also improves predictive performance across many datasets, where various forms of domain knowledge (e.g., feature importance, clinical rules, fairness constraints) are available. Notably, our approach outperforms BNNs that use an uninformative isotropic Gaussian prior, as well as those with more specialized—yet unable to flexibly incorporate such general forms of domain knowledge—priors.

We also present various techniques, based on maximum mean discrepancy (Gretton et al., 2012) and moment matching (with SWAG (Maddox et al., 2019)), for *transferring* a learned domain knowledge prior across different model architectures to increase their overall utility. In general, a BNN prior is *architecture-specific*, i.e., we cannot directly use a prior learned for one BNN in another (e.g., with a different number of hidden layers or units). While relearning a new prior every time is one option, such an approach can be expensive and even infeasible in scenarios when we *no longer have access to the loss* $\phi$. For example, clinical rules derived from patient data in one hospital[1] may not be accessible in another due to privacy. Our empirical results demonstrate that we can efficiently transfer our priors to different model classes, where models sampled from a transferred prior achieve significantly lower values of $\phi$ compared to models drawn from an isotropic Gaussian prior.

To summarize, our contributions are as follows:

---

[1]See **Thresholds Used for Defining** $\phi_{\text{clinical}}$ in Appendix G.3.

1. We propose a novel approach to incorporating *general* forms of domain knowledge (e.g., fairness, clinical rules) that can be specified via a loss function into a prior for BNNs.

2. We propose a variational inference approach that leverages unlabeled data to learn our domain knowledge prior, which is amenable to efficient posterior inference and sampling.

3. We demonstrate that in a semi-supervised learning setup, using our informative prior leads to improved downstream performance and alignment with domain knowledge over commonly used BNN priors (e.g., isotropic Gaussian, GP) on real-world datasets from various domains.

4. We present a strategy for *transferring* a learned informative prior across different neural network architectures, by matching the moments of the learned prior or by maximum mean discrepancy (MMD).

## 2   RELATED WORK

**Learning with Domain Knowledge.**   Many researchers have focused on incorporating domain knowledge or explanations into increasingly black-box deep learning models. Some approaches directly regularize models to incorporate instances of such domain knowledge (Ross et al., 2017; Rieger et al., 2020; Ismail et al., 2021). For example, Rieger et al. (2020) discourage models from using spurious patches in images for skin cancer detection tasks by penalizing models that place high feature importance on those patches. However, prior knowledge can also come in various forms beyond explanations, including rules from physics (de Avila Belbute-Peres et al., 2018; Seo et al., 2021), weak supervision (Sam & Kolter, 2022), invariance (Chen et al., 2020), explicit output constraints for particular regions of the input space (Yang et al., 2020) or desirable properties such as fairness (Zafar et al., 2017; Dwork et al., 2012). These works suggest that incorporating domain knowledge can lead to models that are more robust and perform better out-of-distribution. Existing work theoretically analyzes such simple incorporation of domain knowledge as constraints to show benefits in sample complexity (Pukdee et al., 2023).

In the Bayesian setting, existing works have studied directly regularizing posterior samples (Zhu et al., 2014; Huang et al., 2023), but none have extensively studied how to obtain BNN priors that incorporate forms of domain knowledge as broad as those aforementioned. Moreover, using informative priors is more computationally efficient than posterior regularization, as we only need to compute $\phi$ during the pretraining phase for the prior (see Section 4.2) and not for every posterior sample. Our method thus scales better when sampling a large number of posterior samples, allowing a more accurate approximation of the model posterior average.

**Priors in Bayesian Neural Networks.**   As discussed above, the high-dimensionality of the weight space for BNNs makes specifying a prior that reflects aforementioned forms of domain knowledge challenging (Nalisnick, 2018; Fortuin, 2022). Prior works propose to encode functional properties such as smoothness and periodicity by using a function-space prior (e.g., via GPs (Rasmussen & Williams, 2005)) (Sun et al., 2017; 2019; Hafner et al., 2019; Tran et al., 2022), to encode output constraints for particular regions of the input space into a weight-space prior (Yang et al., 2020), or to use a set of reference models (e.g., simpler linear models) as priors to regularize the predictive complexity of BNNs (Nalisnick et al., 2021). Our work differs from prior work in that we address more general notions of domain knowledge, such as feature importance and fairness, which is difficult to achieve with existing methods (e.g., how does one encode notions of fairness into a GP kernel?).

More recent works propose to leverage advances in self-supervised learning (Henaff, 2020; Chen et al., 2020) to learn more informative and expressive priors from auxiliary, unlabeled data. Sharma et al. (2023b) propose to learn an informative prior by fixing the parameters of the base encoder to the approximate maximum a posteriori (MAP) estimate from contrastive learning (Chen et al., 2020). Shwartz-Ziv et al. (2022) propose to use a temperature-scaled posterior from a source task as a pretrained, informative prior for the target task, and empirically demonstrate that a BNN with an informative prior consistently outperforms BNNs with uninformative priors (e.g., isotropic Gaussian) and non-Bayesian neural network ensembles in predictive accuracy, uncertainty estimation, and data efficiency. Other works look at the usage of priors incorporating knowledge from transfer learning (Lee et al., 2024; Lim et al., 2024). We remark that, to the best of our knowledge, there are no other existing BNN methods that allow incorporating general forms of domain knowledge into BNN priors. The most relevant prior work by Yang et al. (2020), which encodes information by upweighting

models that satisfy particular constraints on their output space, can be seen as a specific instance of our framework but is not easily applicable to most tasks considered in this paper.

# 3 PRELIMINARIES

We consider a standard semi-supervised learning setting. Let $\mathcal{X}$ be an instance space and $\mathcal{Y}$ be a label space. Let $\mathcal{D}$ be a distribution over $\mathcal{X} \times \mathcal{Y}$. We observe a training dataset of examples $X = \{(x_1, y_1), ..., (x_n, y_n)\}$ and unlabeled examples $X' = \{x'_1, ..., x'_k\}$ drawn from $\mathcal{D}$ and a marginal distribution $\mathcal{D}_{\mathcal{X}}$, respectively. We consider a class of neural networks $\mathcal{H} = \{h_w | h_w : \mathcal{X} \to \mathcal{Y}\}$, which have weights $w$. Our goal is to learn a neural network $h$ (or a distribution over possible neural networks with a mean) that achieves the lowest loss, or

$$\text{err}(h) := \mathbb{E}_{(x,y) \sim \mathcal{D}}[\ell(h_w(x), y)],$$

where $\ell$ is the 0-1 loss for classification, and the $\ell_1$ or $\ell_2$ loss for regression.

One approach to capture model uncertainty is via BNNs, which models a distribution over neural networks via a distribution over *weights*, $q(w)$. In practice, it is common to assume a standard isotropic Gaussian prior $q(w) = \prod_i \mathcal{N}(w_i; 0, \sigma_i^2)$. This does not capture any prior knowledge about downstream tasks but is primarily used for its computational tractability. Given a prior $q(w)$ over neural network weights $w$ and labeled data $X$, we can sample from the posterior distribution using stochastic gradient Markov chain Monte Carlo methods such as Stochastic Gradient Hamiltonian Monte Carlo (SGHMC) (Chen et al., 2014) and Stochastic Gradient Langevin Dynamics (SGLD) (Welling & Teh, 2011). In this work, we mainly use SGLD in our experiments (Sections 5.1–5.2) but also consider MultiSWAG (Wilson & Izmailov, 2020) in our ablations (Section 5.3).

# 4 DOMAIN KNOWLEDGE PRIORS FOR BAYESIAN NEURAL NETWORKS

While existing methods tackle specific desirable properties of a network (e.g., smoothness), it is unclear how to incorporate very general forms of domain knowledge into BNNs, as discussed in Sections 1–2. We propose to achieve this by incorporating such information into a data-driven prior.

## 4.1 DOMAIN KNOWLEDGE LOSS

First, we define our notion of domain knowledge. We propose to represent this as a loss function that measures the alignment of a particular model to our domain knowledge.

**Definition 1** *(Domain Knowledge Loss) A domain knowledge loss function can be expressed as $\phi : \mathcal{H} \times \mathcal{X} \to \mathbb{R}$, which takes inputs $h \in \mathcal{H}, x \in \mathcal{X}$ and has $\phi(h, x) \geq 0$.*

We capture how well $h$ satisfies our domain knowledge at a point $x$ through this loss function, where a lower loss value implies that $h$ better satisfies the domain knowledge. This definition is quite general, and it is possible to define the loss $\phi$ to capture various notions of domain knowledge including physical rules and information about spurious correlations (see examples of these losses in Section 5.1). We remark that these notions of domain knowledge are functions of the random input data $x$, and thus are difficult to directly encode in function space or via a kernel in a GP prior.

Given this definition of domain knowledge, we want our models to achieve low values of this loss, e.g., $\mathbb{E}_{x \sim \mathcal{D}_{\mathcal{X}}}[\phi(h, x)] \leq \tau$, where $\tau$ is some threshold. We remark that this loss can be evaluated solely on unlabeled data, which yields nicely to using this for pretraining or learning priors. Considering losses that use information about labels could be potentially interesting, especially in the case of certain fairness metrics, e.g., equal odds and disparate impact (Hardt et al., 2016; Mehrabi et al., 2021).

In the frequentist setting, we can incorporate such domain knowledge by simply adding a regularization term based on this surrogate loss (Ross et al., 2017; Rieger et al., 2020; Pukdee et al., 2023). For a loss function $\ell$, this yields the regularized objective given by

$$\min_{h \in \mathcal{H}} \frac{1}{n} \sum_{i=1}^{n} \ell(h, x_i, y_i) + \lambda \cdot \frac{1}{k} \sum_{i=1}^{k} \phi(h, x'_i), \tag{1}$$

where $\lambda > 0$ is the regularization coefficient. Augmented Lagrangian approaches like Equation 1 can achieve good supervised performance while minimizing the surrogate loss. A similar approach can be taken in the Bayesian case using posterior regularization (Zhu et al., 2014), although we focus the scope of this paper on learning informative priors.

### 4.2 LEARNING INFORMATIVE PRIORS

We present our method for incorporating domain knowledge in the form of these losses into an informative prior for BNNs. As we want to encourage sampling models that achieve low values of the surrogate loss $\phi$, our goal is to learn a prior that assigns high probability mass to these models, consequently influencing samples from the posterior.

We propose to learn our informative prior by inferring the posterior distribution over $w$ given unlabeled data $X'$ and a surrogate loss $\phi$. By Bayes' rule, this posterior is given by

$$p(w|X', \phi) \propto p(\phi|w, X') \cdot p(w),$$

where the weight-space prior $p(w) = \prod_i \mathcal{N}(w_i; 0, \sigma_i^2)$ is the commonly used isotropic Gaussian distribution. Since our goal is to enforce $\phi(h_w, x)$ to be small, we assume that the likelihood for $\phi$ is given by

$$p(\phi|w, x) = \mathcal{N}\left(\phi(h_w, x); 0, \tau^2\right),$$

where $\tau > 0$ is a hyperparameter controlling how much probability mass we want to center about models that most satisfy our domain knowledge. The posterior distribution, which represents our domain knowledge-informed prior that can be used in later tasks, is then given by

$$p(w|X', \phi) \propto \prod_{x'_i \in X'} \mathcal{N}(\phi(h_w, x_i); 0, \tau^2) \cdot p(w). \tag{2}$$

As computing the true posterior in Equation 2 is intractable, we use variational inference (Kingma & Welling, 2013; Blei et al., 2017) to approximate it with the low-rank multivariate Gaussian distribution

$$q_\psi(w) = \mathcal{N}(w; \mu, \Sigma_r), \quad \Sigma_r = \sum_{i=1}^r v_i v_i^T + \sigma^2 I, \tag{3}$$

where $\psi = (\mu, v_1, \ldots, v_r)$ and where $\sigma > 0$ is a small, fixed value that keeps $\Sigma_r$ positive definite and $\mu$ is a vector of real-valued means. We assume that the variational covariance matrix $\Sigma_r$ has low rank $r$ for computational efficiency, given that $w$ is generally high-dimensional, which is a standard assumption in practice. As such, the size of our BNN scales as $O(r \cdot n)$, where $n$ represents the number of parameters in the neural network architecture.

**Our Variational Objective.** We optimize the variational parameters $\psi$ to maximize the evidence lower bound (ELBO) which is given by

$$\mathbb{E}_{w \sim q_\psi}[\log p(\phi|w, X')] - \mathrm{KL}(q_\psi(w)||p(w)). \tag{4}$$

This is a lower bound of $\log p(\phi|X')$, and optimality is achieved when $q_\psi(w) = p(w|\phi, X')$. Since $q_\psi(w)$ and $p(w)$ are both multivariate Gaussian distributions, sampling from these distributions is straightforward, and the KL divergence term between $p(w)$ and $q_\psi(w)$ admits a closed form that can be computed efficiently. Given a set of unlabeled examples $X'$, we thus seek to optimize the objective

$$\max_\psi \left( \mathbb{E}_{w \sim q_\psi} \left[ -\sum_{i=1}^k \frac{\phi(h_w, x)^2}{2\tau^2} \right] - \mathrm{KL}(q_\psi(w)||p(w)) \right).$$

We note that as $\tau \to \infty$, we recover $q_\psi(w) = p(w)$. We reparameterize $\tau$ into $\beta_{\text{pretrain}}$ and rewrite the objective as

$$\max_\psi \left( \mathbb{E}_{w \sim q_\psi} \left[ -\sum_{i=1}^k \phi(h_w, x)^2 \right] - \beta_{\text{pretrain}} \cdot \mathrm{KL}(q_\psi(w)||p(w)) \right).$$

The $\beta_{\text{pretrain}} > 0$ hyperparameter controls the strength of the regularization towards the isotropic Gaussian prior $p(w)$ in our objective. We then use the learned intermediate posterior distribution $q_\psi(w)$ as our informative prior for downstream tasks, performing posterior sampling via methods commonly used in practice (e.g., SGLD, MultiSWAG). Since our informative prior $q_\psi(w)$ is a low-rank Gaussian distribution, we remark that the computational overhead of approximate inference with the informative prior is similar to that of using an isotropic Gaussian prior.

## 4.3 TRANSFERRING INFORMATIVE PRIORS

A key limitation of the learned priors is that they are architecture-specific. To make them usable for downstream tasks where other model architectures may be more suitable, it is important to identify effective techniques for *transferring* these learned priors. Our proposed strategy is to match functions drawn from the learned informative prior and a target prior distribution for the new model architecture.

Formally, let $\mathcal{H}_1 = \{h_w \mid h_w : \mathcal{X} \to \mathcal{Y}\}$ represent the hypothesis class of our original model architecture, with a corresponding informative prior $q_{\psi_1}(w)$. We want to learn an informative prior for a different class of networks $\mathcal{H}_2 = \{h_u \mid h_u : \mathcal{X} \to \mathcal{Y}\}$. We hope to learn a distribution $q_{\psi_2}(u)$ such that the distributions over $\mathcal{H}_1$ and $\mathcal{H}_2$ induced by $w \sim q_{\psi_1}(w)$ and $u \sim q_{\psi_2}(u)$ are close. As we consider low-rank Gaussian priors, we can efficiently draw samples from $q_{\psi_1}(w)$ and $q_{\psi_2}(u)$, and this motivates us to learn $\psi_2$ such that the set of functions $\{h_{w_1}, \ldots, h_{w_n}\} \subseteq \mathcal{H}_1$ and $\{h_{u_1}, \ldots, h_{u_n}\} \subseteq \mathcal{H}_2$ are *similar* when $w_i \sim q_{\psi_1}(w)$, $u_i \sim q_{\psi_2}(u)$. Since the members of each set are functions, it is difficult to compare them directly. If we have access to a set of unlabeled examples $X' = \{x_1', \ldots, x_m'\}$, we can instead make sure that the evaluation of each function on $X'$ are similar, i.e., $W := \{h_{w_1}(X'), \ldots, h_{w_n}(X')\}$ and $U := \{h_{u_1}(X'), \ldots, h_{u_n}(X')\}$ are similar when $h(X') = (h(x_1), \ldots, h(x_m)) \in \mathbb{R}^m$.

**Moment Matching.** We consider simple approaches to match the moments of the two distributions $q_{\psi_1}(w)$ and $q_{\psi_2}(u)$, whose objectives are given by

$$\hat{M}_1 = \mathbb{E}_x[(\mathbb{E}_{w \sim q_{\psi_1}(w)}[h_w(x)] - \mathbb{E}_{u \sim q_{\psi_2}(w)}[h_u(x)])^2]$$

$$\hat{M}_2 = \mathbb{E}_x[(\mathbb{E}_{w \sim q_{\psi_1}(w)}[h_w(x)^2] - \mathbb{E}_{u \sim q_{\psi_2}(w)}[h_u(x)^2])^2],$$

where $\hat{M}_1$ is used to match only the first moment, and $\hat{M}_2$ is used to match the first two moments.

**Maximum Mean Discrepancy.** We propose to minimize the kernel maximum mean discrepancy (MMD) (Gretton et al., 2012; Li et al., 2015) between $W$ and $U$, where the objective is given by

$$\hat{M}(W, U) = \frac{1}{n(n-1)} \sum_{i=1}^{n} \sum_{j \neq i} k(h_{w_i}(X'), h_{w_j}(X')) + \frac{1}{n(n-1)} \sum_{i=1}^{n} \sum_{j \neq i} k(h_{u_i}(X'), h_{u_j}(X'))$$

$$+ \frac{1}{n^2} \sum_{i=1}^{n} \sum_{j=1}^{n} k(h_{w_i}(X'), h_{u_j}(X')),$$

and $k$ represents a kernel. MMD only requires access to the samples from each distribution which fit well with our scenario as these samples are easy to draw. Meanwhile, we remark that other approaches, such as learning $\psi_2$ to fool a discriminator network that is trained to distinguish between two set of samples (Goodfellow et al., 2014; Radford et al., 2016; Arjovsky et al., 2017; Li et al., 2017; Bińkowski et al., 2018) or directly working with kernel two-sample tests for functional data (Wynne & Duncan, 2022), can also be used. A main benefit of studying these prior transferring approaches is that they enable transferring domain knowledge when we no longer have access to the function $\phi$. This approach can help support the open-source release and usage of informative priors, similar to how pretrained models are currently used in practice.

## 5 EXPERIMENTS

We compare our method of learning an informative prior through variational inference, which we refer to as **Banana**, against BNN implementations with various priors, including (1) a standard isotropic Gaussian, (2) a Gaussian with hyperparameters optimized via empirical Bayes using Laplace's method (Daxberger et al., 2021), and (3) a prior that is learned to match a GP prior with a RBF kernel (Tran et al., 2022). We note that the baseline matched to a GP prior (single-output) is not evaluated on our regression dataset (Pendulum), which has multivariate outputs. We also compare against the approximate Bayesian inference method of MC-dropout (Gal & Ghahramani, 2016) and deep ensembles (Lakshminarayanan et al., 2017).

For all prediction tasks described below, we consider a two-layer feedforward neural network with ReLU activations and use SGLD (Welling & Teh, 2011) for posterior inference with the learned

Table 1: Comparison of Banana (with posterior averaging over logits) against BNNs with different priors in terms of accuracy, AUROC, or $L_1$ loss and $\phi$ ($\pm$ s.e.), when averaged over 5 seeds. $\uparrow$ denotes that higher is better, and $\downarrow$ denotes that lower is better. We bold the method with the best performance and the lowest value of $\phi$. - denotes that the corresponding method is not applicable.

| | DecoyMNIST | | MIMIC-IV | | Pendulum | |
|---|---|---|---|---|---|---|
| Method | Accuracy ($\uparrow$) | $\phi_{\text{background}}$ | AUROC ($\uparrow$) | $\phi_{\text{clinical}}$ | $L_1$ Loss ($\downarrow$) | $\phi_{\text{energy\_damping}}$ |
| BNN + Isotropic | $69.05 \pm 1.28$ | $3.35 \pm 0.10$ | $0.6557 \pm 0.0101$ | $0.2910 \pm 0.0070$ | $\mathbf{0.010 \pm 0.002}$ | $0.137 \pm 0.025$ |
| BNN + Laplace | $54.81 \pm 5.21$ | $16.76 \pm 0.95$ | $0.4519 \pm 0.0392$ | $0.2228 \pm 0.0471$ | $21.21 \pm 15.61$ | $0.240 \pm 0.039$ |
| BNN + GP Prior | $71.1 \pm 1.08$ | $3.11 \pm 0.11$ | $0.6563 \pm 0.0102$ | $0.2890 \pm 0.0073$ | - | - |
| **Banana** | $\mathbf{73.63 \pm 0.86}$ | $\mathbf{1.65 \pm 0.05}$ | $\mathbf{0.6778 \pm 0.0026}$ | $\mathbf{0.1924 \pm 0.0047}$ | $\mathbf{0.010 \pm 0.001}$ | $\mathbf{0 \pm 0}$ |

informative prior $q_\psi(w)$ from Equation 3. Meanwhile, we note that the scaling of the prior in SGLD can have a significant impact on downstream predictive performance (Shwartz-Ziv et al., 2022) as well as the weighting of our domain knowledge. As such, we add a hyperparameter $\beta > 0$ that scales the KL divergence term in SGLD, to control the trade-off between using prior information and fitting the observed labeled data. In computing the posterior averages for each method, we average in the logit space of the posterior samples. We explore averaging in the output space of posterior samples in Appendix E.1. For our semi-supervised setting, we use 50% of the original data as our unlabeled data, and 50 labeled examples from each class. We provide additional experimental details in Appendix F.

## 5.1 DATASETS AND DOMAIN KNOWLEDGE LOSSES

**Fairness in Hiring Decisions.** We demonstrate that our method can incorporate notions of fairness on the **Folktables** dataset (Ding et al., 2021). We consider the task of determining whether a particular applicant gets employed, within the Alabama subset of the data in 2018. We focus on group fairness as our underlying domain knowledge, where we define our $\phi$ as

$$\phi_{\text{group\_fairness}}(h, x) = \left(p(h(x)|A = a) - p(h(x)|A = b)\right)^2,$$

where $A$ denotes a random variable for a particular group, such as race or gender. In our experiments, we consider $A = a$ to be the subgroup that corresponds to Black people and $B = b$ to correspond to White people. We note that satisfying this domain knowledge does not necessarily improve predictive performance (Dutta et al., 2020), although it is a desirable and potentially legal necessity of a model.

**Feature Importance for Image Classification.** We also demonstrate that our method can incorporate notions of feature importance. We consider the task of ignoring background information, which are spurious features, on the **DecoyMNIST** dataset (Ross et al., 2017), a variant of MNIST (LeCun et al., 1998). On this task, a patch has been added in the background that correlates with different labels at train and test time. Thus, models that learn to rely on these spurious features for prediction can perform poorly at test time due to such distribution shift. Here, we consider the domain knowledge of ignoring background pixels in making predictions, which can be expressed as

$$\phi_{\text{background}}(h, x) = ||\nabla_x h(x)||_b^2,$$

where $b$ denotes the feature indices that correspond to the background. On DecoyMNIST, we access these feature dimensions by looking at the uncorrupted data, which is not used during training. For other tasks, we can generate these background masks via a segmentation network.

**Clinical Rules for Healthcare Interventions.** We demonstrate how our method can be used to incorporate clinical rules into the prior using the **MIMIC-IV** dataset (Johnson et al., 2023). We reproduce the binary classification task in Yang et al. (2020), where the goal is to predict whether an intervention for hypotension management (e.g., vasopressors) should be given to a patient in the intensive care unit, given a set of physiological measurements. As in Yang et al. (2020), we incorporate the clinical knowledge that an intervention should be made if the patient exhibits: (i) high lactate and low bicarbonate levels, or (ii) high creatinine levels, high blood urea nitrogen (BUN) levels, and low urine output. We can express this knowledge as

$$\phi_{\text{clinical}}(h, x) = \mathbf{1}[x \in \mathcal{X}_c] \cdot \text{ReLU}(1 - h(x)),$$

where $h(x)$ is the classifier output, $\mathcal{X}_c \subseteq \mathcal{X}$ denotes the subset of the input space that satisfies the conditions specified in the above rules, and $\text{ReLU}(1 - h(x))$ encourages $h(x)$ on such inputs to be close to 1. We include all details on cohort selection, data preprocessing, and $\mathcal{X}_c$ in Appendix G.3.

**Physics Rules for Pendulums.** We demonstrate how our method can incorporate physical knowledge into the prior on the **double pendulum** dataset (Seo et al., 2021; Asseman et al., 2018). We consider a regression task where the goal is to predict the next state of double-pendulum dynamics with friction from a given initial state $x = (\theta_1, \omega_1, \theta_2, \omega_2)$ where $\theta_i, \omega_i$ are the angular displacement and the velocity of the $i$-th pendulum, respectively. We incorporate physics knowledge from the law of conservation of energy; since the system has friction, the total energy of the system must be strictly decreasing over time. We can express this knowledge by the following loss:

$$\phi_{\text{energy\_damping}}(h, x) = \max(E(h(x)) - E(x), 0),$$

where $h(x)$ is the predicted next state and $E(x)$ is a function that maps a given state $x$ to its total energy. This loss penalizes predictions of states with higher total energy.

## 5.2 RESULTS

We present the results comparing Banana to baselines of BNNs with other priors in Table 1. We observe that incorporating domain knowledge leads to better-performing classifiers than standard BNN approaches with existing techniques to specify priors. We first note that across all tasks, the model averages produced by Banana achieve lower values of $\phi$ than other baselines. We also remark that the performance of Banana matches or outperforms the other baselines on all tasks, demonstrating the benefits of our approach to incorporate domain knowledge via informative priors.

On the Folktables dataset, $\phi_{\text{group\_fairness}}$ may be at odds with the underlying accuracy, i.e., a less performant model may achieve a lower value of $\phi$ (Pleiss et al., 2017). As such, we present results on this dataset by comparing the trade-offs between accuracy and group fairness observed by each method. In Figure 2, we visualize the density of the posteriors defined by Banana and a BNN with an isotropic Gaussian. We observe that the posterior defined through Banana is more accurate while achieving lower values of group fairness.

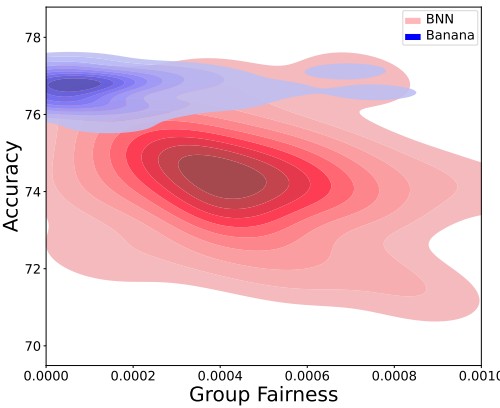

Figure 2: Visualization of the density of the posteriors defined by Banana and a BNN with a standard isotropic Gaussian prior. We have generated these kernel density plots via 50 posterior samples.

### 5.2.1 DIRECTLY SAMPLING FROM THE INFORMATIVE PRIOR

To analyze how well the informative prior encodes our domain knowledge, we can directly sample from our informative prior and compute the value of domain knowledge loss $\phi$ achieved on our sample (see the first two rows of Table 2), although we note that these models are not necessarily suited for a downstream task. We use the same hyperparameter values for training our informative prior as those selected for the downstream classification/regression task in Table 1. On each dataset, we compute our expected value of $\phi$ over 10 samples from the informative prior. We observe that across almost every task, our informative prior successfully upweights models that achieve significantly lower values of $\phi$ on their respective datasets when compared to randomly sampling from an isotropic Gaussian distribution, reflected by a posterior average that has much smaller values of $\phi$.

### 5.2.2 TRANSFERRING PRIORS TO DIFFERENT ARCHITECTURES

In addition to comparing the value of $\phi$ achieved by models sampled from our informative prior, we also evaluate the performance of transferring this informative prior to a different model architecture (see the bottom three rows in Table 2). We transfer the prior over a two-layer neural network to

Table 2: Our proposed methods for transferring priors successfully improve alignment to domain knowledge in BNNs with different architectures. (Top 2 rows) $\phi$ values for models drawn from an isotropic Gaussian prior and the learned Banana prior. (Bottom 3 rows) $\phi$ values for models with larger architectures drawn from an isotropic Gaussian prior and a prior transferred from Banana via MMD or first moment matching (with SWAG). We show the average and standard error over 5 seeds.

| Method | DecoyMNIST | Folktables | MIMIC-IV | Pendulum |
|---|---|---|---|---|
| Isotropic | $0.2499 \pm 0.0199$ | $0.0199 \pm 0.0027$ | $0.2788 \pm 0.0129$ | $135.45 \pm 4.16$ |
| **Banana** | $\mathbf{0.0541 \pm 0.0176}$ | $\mathbf{0.0052 \pm 0.0005}$ | $\mathbf{0.0298 \pm 0.0059}$ | $\mathbf{0.019 \pm 0.017}$ |
| Isotropic (L) | $0.4950 \pm 0.0245$ | $0.0193 \pm 0.0015$ | $0.2986 \pm 0.0066$ | $189.74 \pm 7.46$ |
| **Banana + MMD** | $\mathbf{0.2771 \pm 0.0323}$ | $0.0172 \pm 0.0014$ | $0.0148 \pm 0.0002$ | $0.047 \pm 0.047$ |
| **Banana + 1st Moment (SWAG)** | $0.3419 \pm 0.0051$ | $\mathbf{0.0021 \pm 0.0012}$ | $\mathbf{0.0032 \pm 0.0013}$ | $\mathbf{0.0 \pm 0.0}$ |

another two-layer network with a larger hidden dimension size, where we minimize the difference between first moments (via SWAG (Maddox et al., 2019)) or MMD with respect to a Gaussian kernel. We observe that our transferring approach yields informative priors over the new model class that also reflect much smaller values of $\phi$ than standard isotropic Gaussian prior over the larger model architecture, almost fully recovering the same performance as the original prior in many cases. We further compare against other strategies that we propose to transfer this prior in additional ablations in Appendix E.4. These results demonstrate that informative priors learned in Banana can effectively be transferred to different model architectures that may be better suited for the downstream task.

## 5.3 ABLATIONS

**Alternative Approximations for the Informative Prior.** We assess different approximation schemes for learning the informative prior to better capture the knowledge in $\phi$. Given that variational inference often underestimates the variance of the true posterior (Blei et al., 2017), which need not be unimodal, we consider approximating Equation 2 with a *mixture* $q(w) = \frac{1}{K} \sum_{k=1}^{K} \mathcal{N}(\mu_k, \Sigma_{r,k})$ of $K$ rank-$r$ Gaussians, via the MultiSWAG method (Wilson & Izmailov, 2020). For each $k = 1, \ldots, K$, we initialize the model parameters with a different random seed and compute $(\mu_k, \Sigma_{r,k})$ by averaging over 10 samples (5 epochs apart) from the stochastic gradient descent (SGD) trajectory, after an initial 20 epochs of warmup training. Using each $q_k$ as an informative prior, we sample 5 weights from the downstream posterior via SGLD.

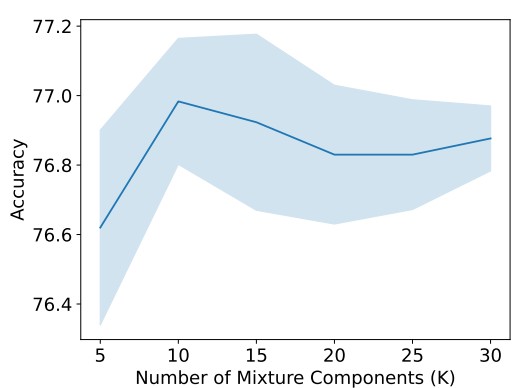

Figure 3: Change in test accuracy on the DecoyMNIST task when varying the number of mixture components in the informative prior in Banana. Results are averaged over 5 seeds, and the shaded region represents mean $\pm$ standard error.

Figure 3 shows the change in test accuracy on DecoyMNIST as we vary $K$ from 5 to 30 in increments of 5. We find that on average, the test accuracy tends to increase with increasing $K$ and improves over the test accuracy (Table 1), before plateauing after a certain complexity of the prior approximation. These results suggest that with a sufficient computational budget, learning a multimodal informative prior can be an effective approach for better capturing our domain knowledge and improving downstream performance. This shows that a sufficiently complex $q$ is required to fully reap the benefits of using prior information. We study this further in Appendix E.3, where we compare with lower and higher rank approximations of our prior.

**Amount of Labeled Data.** We study the benefits of Banana over other BNN alternatives as we vary the amount of labeled data used for sampling from the posterior in Figure 4. In many cases, the domain knowledge can provide information that can be learned from the data; thus, Banana often more strongly outperforms baselines when there is insufficient data to learn this domain knowledge, e.g., Banana strongly outperforms the baseline with 5 data on Pendulum (Figure 4; right). For the

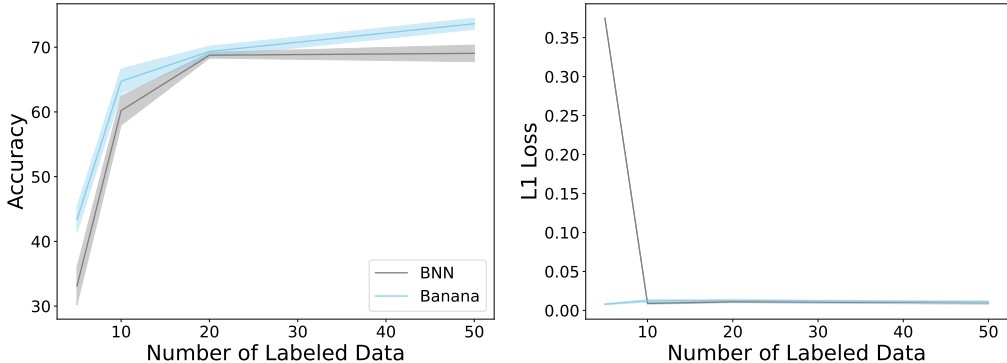

Figure 4: Performance of Banana and a BNN with an isotropic Gaussian prior on DecoyMNIST (left) and Pendulum (right) as we vary the amount of labeled data used in sampling from the posterior. Shaded error regions represent the standard error, computed over 5 seeds.

| Method | MNIST | MIMIC | ACS |
|---|---|---|---|
| BNN + Isotropic | $0.24 \pm 0.01$ | $0.27 \pm 0.01$ | $0.0053 \pm 0.0007$ |
| Banana (ours) | $0.19 \pm 0.00$ | $0.27 \pm 0.01$ | $0.0049 \pm 0.0009$ |

Table 3: Expected calibration error of Banana compared to a BNN + Isotropic Gaussian prior on our considered classification datasets. Results are averaged over 5 seeds.

case of DecoyMNIST (Figure 4; left), the domain knowledge cannot be learned from the data, and the performance improvements of Banana remain across all amounts of labeled data. We defer results on the remaining datasets to Appendix E.7.

## 5.4 Uncertainty Quantification Results

As one of the primary uses of BNNs and Bayesian methods is in uncertainty quantification, we provide an experiment to assess the calibrations (via the ECE) of Banana to the BNN + Isotropic Gaussian baseline. We observe that Banana achieves slightly better or comparable calibration (in terms of ECE) on all tasks in Table 3, meaning that it better quantifies its uncertainty in its predictions.

## 6 DISCUSSION

We propose a framework to incorporate general forms of domain knowledge into the priors for BNNs. Empirically, we observe that this can improve the performance of BNNs across several tasks with different notions of domain knowledge and leads to models that exhibit desirable properties. In addition, we provide an effective approach to transfer informative priors across model architectures, resolving an existing problem in the literature. Our results provide new insights into incorporating domain knowledge into priors for Bayesian methods, which can be captured by optimizing a learnable approximation through variational inference. As a whole, our results support the development of open-source informative priors that practitioners can incorporate into their various specific use cases to encode desirable model properties without the need to deal with $\phi$ directly, irrespective of the desired model architecture. As such, this supports the foundations for pretraining in the Bayesian setting, by providing a framework to develop and transfer informative priors to new model tasks as desired, similar to pretrained weights or foundation models that are currently released as open-source.

**Reproducibility Statement** All code and instructions necessary to reproduce our results and experiments is publicly available at https://anonymous.4open.science/r/banana-iclr/README.md. Further, we include all experimental details (e.g., hyperparameters) for reproducibility in Appendix F.

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

# A  Experiments with Large Labeled Data

While we focus on the semi-supervised learning setting here with limited labeled data, we also provide results, except where we use a much larger amount of training data.

Table A1: Comparison of Banana (with posterior averaging over logits) against BNNs with different priors in terms of accuracy, AUROC, or $L_1$ loss and $\phi$ ($\pm$ s.e.), when averaged over 5 seeds on larger dataset sizes. $\uparrow$ denotes that higher is better, and $\downarrow$ denotes that lower is better. We bold the method with the best performance and the lowest value of $\phi$. - denotes when a method is not applicable.

| | DecoyMNIST | | MIMIC-IV | | Pendulum | |
|---|---|---|---|---|---|---|
| Method | Accuracy ($\uparrow$) | $\phi_{background}$ | AUROC ($\uparrow$) | $\phi_{clinical}$ | $L_1$ Loss ($\downarrow$) | $\phi_{energy\_damping}$ |
| BNN + Isotropic | $76.41 \pm 0.71$ | $1.06 \pm 0.06$ | $0.6981 \pm 0.0003$ | $0.1624 \pm 0.0005$ | $\mathbf{0.0036 \pm 0.0001}$ | $0.0319 \pm 0.0026$ |
| BNN + Laplace | $76.47 \pm 0.70$ | $1.14 \pm 0.04$ | $0.6980 \pm 0.0002$ | $0.1625 \pm 0.0009$ | $0.0043 \pm 0.0006$ | $0.0367 \pm 0.0100$ |
| BNN + GP Prior | $75.49 \pm 0.70$ | $1.54 \pm 0.04$ | $0.6979 \pm 0.0002$ | $0.1628 \pm 0.0008$ | - | - |
| **Banana** | $\mathbf{78.21 \pm 0.40}$ | $\mathbf{0.44 \pm 0.01}$ | $\mathbf{0.6983 \pm 0.0001}$ | $\mathbf{0.1619 \pm 0.0005}$ | $0.0041 \pm 0.0007$ | $\mathbf{0.0025 \pm 0.0010}$ |

Banana still outperforms the alternatives across all tasks, showing that using informative priors can still help even with larger labeled data amounts (Table A1). As expected, we observe less impact of using an informative prior with a larger number of labeled data, although our method still performs the best. The labeled dataset amounts for each task are given in Table A2.

Table A2: Number of labeled data for each dataset with our larger labeled data experiments.

| Dataset | DecoyMNIST | MIMIC-IV | Pendulum |
|---|---|---|---|
| **Samples** | 30,000 | 10,915 | 9,000 |

# B  Experiments with Larger Model Architectures

We also provide experiments with larger model architectures and datasets to show that our approach still benefits at scale. For the dataset, we focus on the Waterbirds dataset (Sagawa et al., 2019), which consists of bird images from the CUB dataset combined with backgrounds from the Places dataset. The task is a binary classification problem: determining whether an image depicts a waterbird or a landbird. However, there are spurious correlations in the training data, where landbirds predominantly appear against land backgrounds, and waterbirds against water backgrounds.

The goal is to ensure that the model does not rely on background information for making predictions. To evaluate this, we measure the accuracy for each subgroup of background and label and aim to maximize the *worst-group accuracy*. Given the known spurious correlation between image backgrounds and labels, we leverage domain knowledge, similar to the approach used in DecoyMNIST, by penalizing the gradient of the background in the images.

For the model, we use a ResNet-18 architecture. Following Sharma et al. (2023a), which argues that partially stochastic networks can match or even outperform fully stochastic networks, we freeze the backbone of the ResNet-18 and only train the linear head. Our results show that **Banana** achieves better worst-group accuracy compared to a standard BNN. Additionally, Banana exhibits lower input gradient magnitudes across all groups. This demonstrates the effectiveness of our approach at a larger model and dataset scale.

| Method | Accuracy | $\phi_{\mathbf{waterbirds}}$ | Worst-Group Accuracy | Worst-Group $\phi_{\mathbf{waterbirds}}$ |
|---|---|---|---|---|
| BNN + Isotropic | $56.71 \pm 0.32$ | $0.263 \pm 0.017$ | $19.034 \pm 3.964$ | $0.330 \pm 0.022$ |
| **Banana** | $\mathbf{70.34 \pm 0.26}$ | $\mathbf{0.034 \pm 0.001}$ | $\mathbf{42.15 \pm 1.09}$ | $\mathbf{0.037 \pm 0.001}$ |

Table A3: Comparison of Banana and BNN + Isotropic on the Waterbirds dataset.

## C  ADDITIONAL INFORMATIVE PRIOR BASELINES

We provide an additional comparison to the work of Shwartz-Ziv et al. (2022), which also incorporates an informative prior that is learned from a pretrained checkpoint and then fitting a distribution around this learned set of weights with SWAG Maddox et al. (2019). We remark that this approach has demonstrated to work in cases of self-supervised learning, which does not generally admit many degenerate optimal parameter solutions to the optimization problem. However, in many of the cases for our domain knowledge losses $\phi$, this is generally the case and can be a problem with such approaches.

Table A4: Comparison of Banana against the informative prior produced by Pretrain Your Loss (Shwartz-Ziv et al., 2022) in terms of accuracy, AUROC, or $L_1$ loss and $\phi$ ($\pm$ s.e.), when averaged over 5 seeds. $\uparrow$ denotes that higher is better, and $\downarrow$ denotes that lower is better. We bold the method with the best performance and the lowest value of $\phi$. - denotes that the corresponding method is not applicable.

| | DecoyMNIST | | MIMIC-IV | | Pendulum | |
|---|---|---|---|---|---|---|
| Method | Accuracy ($\uparrow$) | $\phi_{\text{background}}$ | AUROC ($\uparrow$) | $\phi_{\text{clinical}}$ | $L_1$ Loss ($\downarrow$) | $\phi_{\text{energy\_damping}}$ |
| Pretrain Your Loss | $71.79 \pm 1.35$ | $\mathbf{1.62 \pm 0.02}$ | $0.6670 \pm 0.0101$ | $0.2012 \pm 0.0074$ | $0.330 \pm 0.003$ | $\mathbf{0 \pm 0}$ |
| **Banana** | $\mathbf{73.63 \pm 0.86}$ | $1.65 \pm 0.05$ | $\mathbf{0.6778 \pm 0.0026}$ | $\mathbf{0.1924 \pm 0.0047}$ | $\mathbf{0.010 \pm 0.001}$ | $\mathbf{0 \pm 0}$ |

We observe that incorporating this domain knowledge via Banana outperforms Pretrain Your Loss across all tasks (Table A4). This shows that indeed our approach is better for the types of domain knowledge losses that we focus on in this paper.

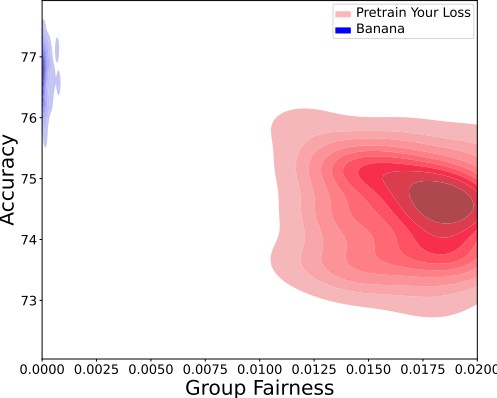

Figure A1: Visualization of the density of the posteriors defined by Banana and a Pretrain Your Loss. We have generated these kernel density plots via 50 posterior samples.

Furthermore, we show added results in terms of fairness, where we again see that the degenerate solutions learned by a single checkpoint lead to the failure of incorporating the fairness constraints on the Folktables dataset for the Pretrain Your Loss baseline (Figure A1).

## D  EXPERIMENTS WITH MISSPECIFIED PRIORS

We also provide an experiment where we use an incorrect inductive bias on the MIMIC-IV task. Instead of using correct thresholds in $\phi$, we compute $\phi$ with the reverse of each threshold (e.g., with low lactate or high bicarbonate levels). This is clearly an incorrect inductive bias, as this does not encode any useful information.

We observe that this as expected hurts performance, although the performance is still comparable with BNNs with the standard isotropic prior. Generally, using broad forms of inductive bias (as what we experiment with in our paper in Section 5.1) do not hurt model performance.

| Dataset | MIMIC AUROC |
|---|---|
| BNN + Isotropic | $69.05 \pm 1.28$ |
| Banana | $73.63 \pm 0.86$ |
| Banana (misspecified) | $71.92 \pm 0.73$ |

Table A5: MIMIC Accuracy with a misspecified Banana prior. We observe that performance does not degrade too much with a misspecified prior.

# E    ADDITIONAL EXPERIMENTS

## E.1    MODEL AVERAGING

We provide experiments to compare averaging different samples from the posterior distribution in their logit space and in their prediction space. We remark that the Pendulum dataset consists of a regression task, where there is no distinction between logit space and prediction space and, thus, we do not report those results as they are the same. We also note that while it is common to take the average in the weight space, this approach performs quite poorly in our setup since we do not control the norms of each layer (i.e., there are no layer normalization operations).

Table A6: A comparison of different ensembling techniques of models sampled from the posterior distribution of Banana. We report accuracy ($\pm$ s.e.) when averaged over 5 seeds.

| | DecoyMNIST | Folktables | MIMIC-IV |
|---|---|---|---|
| Banana - Logits | $73.63 \pm 0.86$ | $75.75 \pm 0.28$ | $0.6778 \pm 0.0026$ |
| Banana - Predictions | $71.93 \pm 0.70$ | $75.81 \pm 0.28$ | $0.6770 \pm 0.0025$ |

We note that there is not a significant difference, although we observe that performing model averaging over the logits of each sample from the posterior distribution achieves slightly higher accuracy than averaging over the discrete predictions (where ties are broken by taking the first class in order).

## E.2    COMPARISON AGAINST BAYESIAN APPROXIMATIONS

To contextualize our results with other common approximations of Bayesian inference in the literature, we present additional comparisons against the standard methods of MC-dropout (Gal & Ghahramani, 2016) and deep ensembles (Lakshminarayanan et al., 2017).

Table A7: A comparison of different ensembling techniques of models sampled from the posterior distribution of Banana. We report accuracy ($\pm$ s.e.) when averaged over 5 seeds.

| | DecoyMNIST | MIMIC-IV | Pendulum |
|---|---|---|---|
| Deep Ensemble | $70.81 \pm 1.36$ | $\mathbf{0.6810 \pm 0.0010}$ | $0.012 \pm 0.002$ |
| MC-Dropout | $69.85 \pm 1.33$ | $0.6685 \pm 0.0037$ | $0.017 \pm 0.004$ |
| **Banana** | $\mathbf{73.63 \pm 0.86}$ | $0.6778 \pm 0.0026$ | $\mathbf{0.010 \pm 0.001}$ |

## E.3    VARYING THE COMPLEXITY OF OUR INFORMATIVE PRIOR APPROXIMATION

As demonstrated in Table 2, our approach can capture domain knowledge in the form of $\phi$ through a rank-$r$ approximation of the covariance matrix of a multivariate Gaussian distribution. Here, we run ablations to study how the rank of our approximation influences downstream performance, albeit while suffering slightly larger computational costs (i.e., $O(rn)$ where $r$ is the rank and $n$ is the number of parameters).

We observe that increasing the rank of our prior approximation on the DecoyMNIST task with 50 labeled data seems to slightly improve performance, with larger rank approximations plateuing in performance after $r = 20$. (Figure A2). This slight increase demonstrates that learning informative priors with strong performance suffices with a small rank approximation, which is not too computationally expensive.

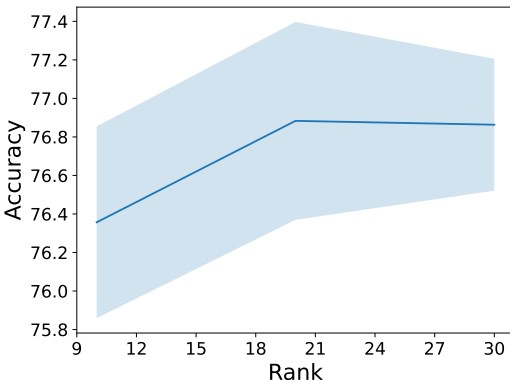

Figure A2: Results when varying the rank to approximate our informative prior in Banana on the DecoyMNIST task, averaged over 5 seeds. Shaded region represents mean $\pm$ standard error.

Table A8: Comparing the value of $\phi$ of models drawn from an isotropic Gaussian prior and an informative prior transferred from Banana to a smaller network size (S) in terms of hidden dimension size via multiple moment matching techniques and MMD. Results are averaged over 5 seeds.

| Method | DecoyMNIST | Folktables | MIMIC-IV | Pendulum |
|---|---|---|---|---|
| Isotropic (S) | $0.1230 \pm 0.0172$ | $0.0199 \pm 0.0027$ | $0.2914 \pm 0.0031$ | $96.84 \pm 1.63$ |
| Banana + 1st Moment | $0.0000 \pm 0.0000$ | $0.0091 \pm 0.0045$ | $0.0141 \pm 0.0036$ | $16.48 \pm 4.55$ |
| Banana + 1st and 2nd Moment | $0.0410 \pm 0.0338$ | $0.0132 \pm 0.0018$ | $0.1592 \pm 0.0192$ | $191.22 \pm 16.76$ |
| Banana + MMD | $0.2400 \pm 0.1050$ | $0.0205 \pm 0.0020$ | $0.0135 \pm 0.0015$ | $0.0214 \pm 0.0200$ |
| Banana + 1st Moment (SWAG) | $1.303 \pm 0.0283$ | $\mathbf{0.0025 \pm 0.0016}$ | $\mathbf{0.0032 \pm 0.0013}$ | $\mathbf{0.0 \pm 0.0}$ |

### E.4 COMPARISON AGAINST OTHER PRIOR TRANSFER TECHNIQUES

We provide comparisons against additional techniques to transfer the prior learned in Banana across different model architectures. We observe that MMD and 1st Moment Matching using SWAG (Maddox et al., 2019) perform favorably when compared to simply matching and directly optimizing over the learnable parameters of the prior approximation.

### E.5 COMPARISON AGAINST FREQUENTIST APPROACHES

While not the main focus of our paper, we also provide a comparison against standard frequentist approaches to incorporate domain knowledge. We compare against a standard supervised learning approach and a **Lagrangian**-penalized approach, where we can directly regularize with the value of $\phi$ times some hyperparameter $\lambda$, as in Eq. equation 1. We also consider an ensemble of such Lagrangian-penalized methods, which we refer to as **Lagrangian ensemble**. We also remark that this would be similar to the performance of posterior regularization.

Table A9: We compare Banana against frequentist analogues that incorporate domain knowledge and report the accuracy, AUROC, or $L_1$ loss and $\phi$ ($\pm$ standard error) when averaged over 5 seeds.

| Method | DecoyMNIST | MIMIC-IV | Pendulum |
|---|---|---|---|
| Lagrangian | $56.90 \pm 6.29$ | $0.6837 \pm 0.0012$ | $2.000 \pm 0.490$ |
| Lagrangian Ens. | $74.43 \pm 0.96$ | $0.6821 \pm 0.0023$ | $0.011 \pm 0.001$ |
| **Banana** | $73.63 \pm 0.86$ | $0.6778 \pm 0.0026$ | $0.010 \pm 0.001$ |

In Table A9, we observe it seems more effective to directly regularize with $\phi$ in making the values of $\phi$ smaller. On the MIMIC dataset, we observe that all methods are comparable. This is likely due to the domain knowledge not being particularly helpful given the amount of labeled data (2000

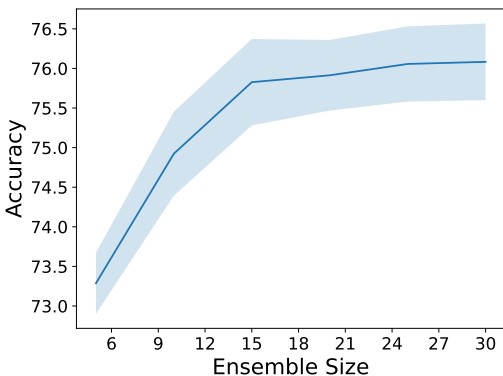

Figure A3: Change in test accuracy on the DecoyMNIST task when varying the number of models sampled to compute our posterior average in Banana. Results are averaged over 5 seeds, and the shaded region represents mean ± standard error.

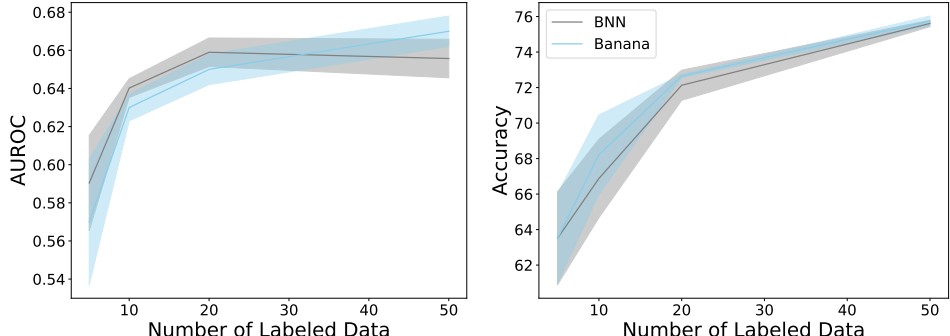

Figure A4: Performance of Banana and a BNN with an isotropic Gaussian prior on MIMIC-IV (Left) and Folktables (Right) as we vary the amount of labeled data used in sampling from the posterior. Shaded error regions represent the standard error, computed over 5 seeds.

examples); this is supported by the observation that the supervised and lagrangian methods have similar performance. However, we again note that performing Lagrangian ensembling methods are more computationally intensive, as it requires regularizing with $\phi$ during each model training process.

### E.6  ABLATIONS ON MODEL ENSEMBLE SIZE

With frequentist methods, computing a model ensemble approximately equivalent to a Bayesian model average (Lakshminarayanan et al., 2017) requires performing potentially computationally expensive regularization with $\phi$ (or even pretraining, where domain knowledge can be incorporated as a notion of invariance for self-supervised learning). With a Bayesian approach on the other hand, we only need to learn the informative prior *once* and can generate multiple samples from the posterior using the learned prior. Given that sampling from the posterior is efficient as in our setting, this can be a significant benefit, specifically when computing this regularization or pretraining is costly.

As such, we run ablation studies to evaluate how the model ensemble's size impacts Banana's downstream performance and the alternative approaches. We observe that increasing the ensemble size increases performance, until we observe diminishing returns after ensemble sizes of 15, on the DecoyMNIST task (Figure A3). This demonstrates that larger ensembles generally achieve better performance and supports the use of informative priors, which can more efficiently scale to posterior averages with larger ensembles when compared to other regularization-based approaches.

### E.7 ADDITIONAL ABLATIONS FOR VARYING LABELED DATA

We present the experiments in varying the amount of labeled data on the MIMIC-IV and Folktables tasks (Figure A4). We observe comparable performance to the BNN with the isotropic Gaussian as we vary the amount of labeled data, and slightly outperform it on the Folktables task.

## F  EXPERIMENTAL DETAILS

**Hyperparameters**  We perform hyperparameter optimization over the following hyperparameter values, selecting the best-performing method on the validation set. For all methods, we consider two-layer neural networks with a ReLU activation function and a hidden dimension size $\in [8, 16, 32]$ for the Folktables dataset, $\in [8]$ for DecoyMNIST and Pendulum, and $\in [32, 64, 96]$ for MIMIC-IV. We also consider batch sizes in $[129, 256, 512]$ for Folktables, $[128, 256]$ for DecoyMNIST and Pendulum, and $[128, 256, 512]$ for MIMIC-IV. We remark that on DecoyMNIST the value of gradients (with respect to input data) is quite sensitive to the overall scale of the for the learnable parameters of the informative prior in Banana. Therefore, we use an initialization randomly sampled from $\mathcal{N}(0, 0.01)$. We also use a $N(0, 0.01)$ initialization for Pendulum. On other tasks, we simply initialize the parameters with $\mathcal{N}(0, 1)$ as they are not as sensitive. For BNNs, we similarly consider a prior distribution of $\mathcal{N}(0, \sigma^2 I)$, where $\sigma^2$ is a hyperparameter tuned on the validation set. For specific methods, we use the following hyperparameters.

**Supervised and Lagrangian**

- learning rate $\in [0.01, 0.001, 0.005, 0.0001]$
- epochs $\in [10, 20, 50, 100]$
- $\lambda \in [1, 0.1, 0.01, 0.001]$
- weight decay $\in [0.1, 0.01, 0]$, used in a standard $L_2$ penalization over network weights

**BNN and Banana**

- number of models (posterior samples) $= 5$
- pretraining epochs $\in [10, 20, 50, 100]$
- posterior epochs $\in [10, 20, 50, 100]$ in MIMIC, Folktables, Pendulum; posterior epochs $\in [5, 10, 15]$ in DecoyMNIST
- $\beta \in [1, 10^{-4}, 10^{-8}, 10^{-12}, 10^{-16}]$
- $\beta_{\text{pretrain}} \in [1, 10^{-4}, 10^{-8}, 10^{-12}, 10^{-16}]$
- prior learning rate $\in [0.1, 0.05, 0.03, 0.01, 0.005, 0.003]$
- posterior learning rate $\in [0.1, 0.01, 0.001]$
- $\sigma^2 \in [1, 0.01]$
- Rank $= 30$

**Compute Resources**  Each experiment was run on a single GeForce 2080 Ti GPU.

## G  DATASET DETAILS

### G.1 DETAILS ON FOLKTABLES

We use the Folktables dataset for the task of determining the employment of a particular job applicant. We restrict our focus to the Alabama subset of the data from 2018. We refer readers to (Ding et al., 2021) for more specific details about the dataset and its collection.

### G.2 DETAILS ON PENDULUM DATASET

On the Pendulum dataset (Seo et al., 2021), we use the configuration detailed in Table C1 for generating the time-series data. We refer the readers to Seo et al. (2021) for full details on the dataset.

Table C1: Constants used in the generation of the Pendulum dataset.

| Dataset Configuration | Value |
|---|---|
| String 1 Length | 1 |
| String 2 Length | 1 |
| Mass 1 | 1 |
| Mass 2 | 5 |
| Friction Coefficient 1 | 0.001 |
| Friction Coefficient 2 | 0.001 |

### G.3 DETAILS ON HEALTHCARE DATA

MIMIC-IV (Johnson et al., 2023) is an open-access database that consists of deidentified electronic health record data collected at the Beth Israel Deaconness Medical Center from 2008 to 2019, covering over 400k distinct hospital admissions. For the intervention prediction task described in Section 5.1, we focus on admissions that include a stay in the intensive care unit (ICU), for which various physiological measurements from bedside monitors, lab tests, etc. are readily available at higher temporal resolution. We provide details on how the study cohort was selected for the experiments, how the features and labels were extracted, and a demographics summary of the final resulting cohort.

**Cohort Selection.** For our study cohort, we include all ICU stays that satisfy the following criteria:

- Adult patients: Given that physiology of young children and adolescents can differ significantly from that of adults, we only include ICU stays corresponding to adult patients between the age of 18 and 89 at the time of admission.

- First ICU stay: Following standard practice (Wang et al., 2020), if a patient had multiple ICU stays across all hospitalizations, we only include the first ICU stay.

- Length of ICU stay $\geq$ 48 hours: We only include ICU stays that lasted long enough to have a sufficient number of measurements for every stay and remove outlier cases.

We note that not all ICU stays selected by this inclusion criteria are eventually included, due to the additional filtering steps detailed in the description on feature and label extraction below. We include a summary of demographic information for the final extracted cohort in Table C2.

**Feature and Label Extraction.** For all ICU stays included in the cohort, we extract the same set of features (2 static and 6 time-dependent features) used in Yang et al. (2020), listed below.

- `Mean Arterial Pressure (MAP)`: Time-Dependent

- `Age at Admission`: Static

- `Urine Output`: Time-Dependent

- `Weight at Admission`: Static

- `Creatinine`: Time-Dependent

- `Lactate`: Time-Dependent

- `Bicarbonate`: Time-Dependent

- `Blood Urea Nitrogen (BUN)`: Time-Dependent

Every recorded time-dependent feature has an associated time stamp (e.g., `2180-07-23 23:50:47`), and we use the measurement time offset from the start time of the corresponding ICU stay to aggregate all measurements into hourly bins and obtain a discretized time-series representation. Within each hourly bin, if more than one measurements are available, we take the most recent measurement. For the static features, we duplicate them along all hourly bins. For example, suppose that a patient's first ICU stay lasted for 2 days. We then obtain a $48 \times 8$ time-series representation, where the rows correspond to the 48 hourly bins, and the columns correspond to the 6 time-dependent features and the 2 static features duplicated along all rows.

Table C2: Summary of demographics for the final extracted cohort of ICU patients. Except for the total number of ICU patients included, we report the mean and standard deviation (in parentheses) of each demographic feature.

|  |  | Missing | Overall |
|---|---|---|---|
| Number of Patients |  |  | 13944 |
| Age |  | 0 | 64.3 (15.7) |
| Gender | Female | 0 | 5751 (41.2) |
|  | Male |  | 8193 (58.8) |
| Ethnicity | Asian | 0 | 394 (2.8) |
|  | Black |  | 1464 (10.5) |
|  | Hispanic |  | 525 (3.8) |
|  | Native American |  | 57 (0.4) |
|  | Other/Unknown |  | 2451 (17.6) |
|  | White |  | 9053 (64.9) |
| Admission Height |  | 3843 | 169.7 (10.5) |
| Admission Weight |  | 0 | 84.0 (25.1) |
| Length of Stay |  | 0 | 185.5 (185.4) |
| ICU Type | Cardiac Vascular ICU (CVICU) | 0 | 2317 (16.6) |
|  | Coronary Care Unit (CCU) |  | 1625 (11.7) |
|  | Medical Intensive Care Unit (MICU) |  | 3432 (24.6) |
|  | Medical/Surgical ICU (MICU/SICU) |  | 2237 (16.0) |
|  | Neuro Intermediate |  | 44 (0.3) |
|  | Neuro Stepdown |  | 16 (0.1) |
|  | Neuro Surgical ICU (Neuro SICU) |  | 386 (2.8) |
|  | Surgical Intensive Care Unit (SICU) |  | 1933 (13.9) |
|  | Trauma SICU (TSICU) |  | 1954 (14.0) |

As in Yang et al. (2020), we consider a *time-independent* binary classification task, where we treat the 8-dimensional features at each hourly bin as a separate sample and predict whether an intervention for hypotension management is necessary for the given hour. To obtain the hourly labels to predict, we extract the start and end times of all recorded vasopressor (e.g., norepinephrine, dobutamine) administrations for each ICU stay, and label each hourly bin as 1 if the vasopressor duration coincides with the hourly bin.

Additionally, given that clinical measurements are measured at different intervals and high levels of sparsity, we filter out all rows that have missing features. Concatenating all samples together, we obtain input features $X \in \mathbb{R}^{49953 \times 8}$ and labels $Y \in \{0, 1\}^{49953}$, where the labels are approximately balanced (positive: 25171 samples, negative: 23565 samples). We then take a stratified 70-15-15 split to get the training, validation, and test datasets while preserving the label proportions, and standardizing all features to zero mean and unit variance based on the training data. We also note that we take a subset of the training data when used to compare all of our approaches; we use a total of 2000 examples.

**Thresholds Used for Defining** $\phi_{\textbf{clinical}}$**.** In adults, the normal range for lactate levels are 0.5–2.2 mmol/L[2] and bicarbonate levels are 22–32 mmol/L[3], and we therefore define the thresholds $\tilde{x}_{\text{lactate}} = 2.2$ and $\tilde{x}_{\text{bicarbonate}} = 22$ and *standardize these values according to the training data.*

---

[2] https://www.ucsfhealth.org/medical-tests/lactic-acid-test
[3] https://myhealth.ucsd.edu/Library/Encyclopedia/167,bicarbonate

