# OpenReview forum: "Bayesian Neural Networks with Domain Knowledge Priors"
_ICLR.cc/2025/Conference — Submitted to ICLR 2025_

### Official Review · Reviewer_Sfqk · 2024-10-27

**Soundness:** 3
**Presentation:** 3
**Contribution:** 3
**Rating:** 6
**Confidence:** 4

**Summary:**

This paper introduces a method for learning an informative prior through variational inference, as an alternative to the commonly used uninformative Gaussian prior distribution in Bayesian Neural Networks (BNNs). And, it proposes a technique for incorporating various inductive biases—difficult to represent with traditional methods—by converting them into a loss form that can be learned as a prior.

**Strengths:**

Strengths

- The paper is well-written and easy to follow.
- Research that enables the efficient learning of human inductive biases or specific constraints in the form of a prior distribution is valuable, and the potential to create priors containing diverse information, such as fairness, is an interesting aspect of this work.

**Weaknesses:**

Weaknesses

- In Section 4, it’s unclear why the authors use $\phi$ as the likelihood when calculating the posterior distribution, rather than directly calculating a posterior that reflects this loss. Instead, they create a prior distribution first and then sample from the posterior using SGMCMC methods. This two-stage sampling approach seems likely to reduce computational efficiency significantly.

- Another drawback is that everything must be represented in the form of a loss. This loss-based formulation can already be interpreted as a regularization term on the likelihood or as a prior distribution in standard MAP solutions or posterior sampling methods, which raises questions about whether this approach is truly novel.

- In addition to [1], which the paper mentions as related work, other studies such as [2] and [3] have also explored methods for creating informative prior distributions. Adding these to the related work section would strengthen the context.

- A key and critical point that prevents me from leaning toward acceptance is that the experiments were conducted only on a simple 2-layer MLP. As model and data scales increase, various methods can behave quite differently, and the experiment on such a limited model scale does not demonstrate that the proposed prior distribution can be effectively applied to larger models.

- The baseline should also include methods like [1] and [2]. Similar to how the authors pre-trained the model using unlabeled data to learn a prior distribution, [1] and [2] also use self-supervised learning to pre-train models, which can then be employed as informative priors.

- To ensure that the proposed method can be safely used for posterior sampling or learning, we need to examine whether it experiences severe performance degradation or operates with a degree of robustness in cases where the inductive bias is incorrect, leading to a misspecified prior distribution, i.e., under model-data misspecification conditions.

References

[1] Shwartz-Ziv, R., Goldblum, M., Souri, H., Kapoor, S., Zhu, C., LeCun, Y., & Wilson, A. G. (2022). Pre-train your loss: Easy bayesian transfer learning with informative priors. Advances in Neural Information Processing Systems, 35, 27706-27715.

[2] Lee, H., Nam, G., Fong, E., & Lee, J. Enhancing Transfer Learning with Flexible Nonparametric Posterior Sampling. In The Twelfth International Conference on Learning Representations.

[3] Lim, S., Yeom, J., Kim, S., Byun, H., Kang, J., Jung, Y., ... & Song, K. (2024). Flat Posterior Does Matter For Bayesian Transfer Learning. arXiv preprint arXiv:2406.15664.

**Questions:**

See Weaknesses section.

---

> ### Author Response · Authors · 2024-11-27
> **Author Response to Reviewer Sfqk (Part 1)**
>
> We thank the reviewer for their efforts in providing a detailed review. We address your concerns below.
>
> > **In Section 4, it’s unclear why the authors use ϕ as the likelihood when calculating the posterior distribution, rather than directly calculating a posterior that reflects this loss. Instead, they create a prior distribution first and then sample from the posterior using SGMCMC methods. This two-stage sampling approach seems likely to reduce computational efficiency significantly.**
>
> We actually believe that this two-stage sampling method is more efficient than incorporating $\phi$ for each posterior sample. We only need to learn the prior a single time, which involves training on $\phi$ once. However, when using domain knowledge at the stage of sampling from the posterior, each sample must be trained with $\phi$. We show that BNNs benefit from averaging multiple a larger number of posterior samples in (Figure A2 in the Appendix), and with these larger posterior averages, one needs to incorporate $\phi$ every time we draw a new sample from the posterior distribution. This is $k$ times more expensive when $k$ is the number of posterior samples. Thus, computational efficiency is actually a benefit of our approach.
>
>
> > **Another drawback is that everything must be represented in the form of a loss. This loss-based formulation can already be interpreted as a regularization term on the likelihood or as a prior distribution in standard MAP solutions or posterior sampling methods, which raises questions about whether this approach is truly novel.**
>
> While these losses could be introduced during posterior sampling, no prior work has attempted to use and incorporate such general forms of domain knowledge into the prior for BNNs. This is a key distinction and is precisely what makes our work novel and a new contribution. As previously mentioned, there are computational efficiency benefits of doing incorporating this information only a single time when learning an informative prior.
>
> > **In addition to [1], which the paper mentions as related work, other studies such as [2] and [3] have also explored methods for creating informative prior distributions. Adding these to the related work section would strengthen the context.**
>
> Thank you for providing these related works. We have added these in our revision. We remark that the key distinction as these two approaches focus on self-supervised learning and the techniques to learn the prior are indeed different.
>
> > **A key and critical point that prevents me from leaning toward acceptance is that the experiments were conducted only on a simple 2-layer MLP. As model and data scales increase, various methods can behave quite differently, and the experiment on such a limited model scale does not demonstrate that the proposed prior distribution can be effectively applied to larger models.**
>
> In response to your request, we have added an additional experiment using a larger model and a larger dataset. For the dataset, we focus on Waterbirds [3], which consists of bird images from the CUB dataset combined with backgrounds from the Places dataset. The task is a binary classification problem: determining whether an image depicts a waterbird or a landbird. However, there are spurious correlations in the training data, where landbirds predominantly appear against land backgrounds, and waterbirds against water backgrounds.
>
> The goal is to ensure that the model does not rely on background information for making predictions. To evaluate this, we measure the accuracy for each subgroup of background and label and want to maximize the worst-group accuracy. Given the known spurious correlation between image backgrounds and labels, we leverage domain knowledge, similar to the approach used in DecoyMNIST, by penalizing the gradient of the background in the images.
>
> For the model, we use a ResNet-18 architecture. Following [2], which argues that partially stochastic networks can match or even outperform fully stochastic networks, we freeze the backbone of the ResNet-18 and only train the linear head.
>
>
> | Metric  | BNN + Isotropic | Banana |
> |-|-|-|
> | **Test Accuracy** | $56.71 \pm 0.32$ |  **70.34 $\pm$ 0.26**  |
> | **Worst-Group Accuracy** |  $19.03\pm 3.96$ | **42.15 $\pm$ 1.09** |
> | **$\phi$**  |  $0.263 \pm 0.017$ | **0.034 $\pm$ 0.001** |
> | **Worst-Group $\phi$**   | $0.330 \pm 0.022$ | **0.037 $\pm$ 0.001** |
>
> Our results show that Banana achieves better accuracy and worst-group accuracy compared to a standard BNN with an isotropic Gaussian prior. Additionally, Banana exhibits lower input gradient magnitudes (lower values of $\phi$) across all data and specifically on the worst-group as well. This demonstrates the effectiveness of our approach and the utility of incorporating domain knowledge through a prior at a larger model and dataset scale.

---

> ### Author Response · Authors · 2024-11-27
> **Author Response to Reviewer Sfqk (Part 2)**
>
> > **The baseline should also include methods like [1] and [2]. Similar to how the authors pre-trained the model using unlabeled data to learn a prior distribution, [1] and [2] also use self-supervised learning to pre-train models, which can then be employed as informative priors.**
>
> We have added a comparison to [1], which also learns an informative prior during pretraining. We remark that [2] does not provide their code, so we cannot compare with their method. We observe the following results:
>
> | Dataset | MNIST Accuracy | MNIST $\phi$ | MIMIC-IV AUROC | MIMIC-IV $\phi$ | Pendulum L1 Loss | Pendulum $\phi$ |
> |-|-|-|-|-|-|-|
> | Pretrain Loss [1] | 71.79 $\pm$ 1.35 | 1.62 $\pm$ 0.02 | 0.6670 $\pm$ 0.0101 | 0.2012 $\pm$ 0.0074 | 0.330 $\pm$ 0.003 | 0.0 $\pm$ 0.0  |
> | Banana | 73.63 $\pm$ 0.86 | 1.65 $\pm$ 0.05 | 0.6778 $\pm$ 0.0026 | 0.1924 $\pm$ 0.0047 | 0.010 $\pm$ 0.001 | 0.0 $\pm$ 0.0 |
>
> We also provide a comparison with this task on the fairness dataset. We provide the accuracy and fairness of the posterior mean ($\pm$ 1 standard deviation). Overall, we find that the posterior mean learned via Banana is much more accurate and fair than the Pretrain Your Loss baseline, due to the likely degenerate solution learned via the original trained model on $\phi$.
>
> | | Accuracy | Group Fairness |
> |-|-|-|
> | Pretrain Your Loss [1] | 74.49 $\pm$ 0.63 | 0.0175 $\pm$ 0.003 |
> | Banana | **76.68 $\pm$ 0.34 | **0.00012 $\pm$ 0.00016** |
>
> Since we cannot provide figures via OpenReview, we present the figure of the whole distirbutions in our revision in Appendix D.
>
> [1] Shwartz-Ziv, et. al. Pre-Train Your Loss: Easy Bayesian Transfer Learning with Informative Priors
>
>
> > **To ensure that the proposed method can be safely used for posterior sampling or learning, we need to examine whether it experiences severe performance degradation or operates with a degree of robustness in cases where the inductive bias is incorrect, leading to a misspecified prior distribution, i.e., under model-data misspecification conditions.**
>
> In response to your request, we provide a new experiment where we use an incorrect inductive bias on the MIMIC-IV task. Instead of using correct thresholds in $\phi$, we compute $\phi$ with the reverse of each threshold (e.g., with low lactate or high bicarbonate levels). This is clearly an incorrect inductive bias, as this does not encode any useful information.
>
> We observe that this as expected hurts performance, although the performance is still comparable with BNNs with the standard isotropic prior. Generally, using broad forms of inductive bias (as what we experiment with in our paper in Section 5.1) do not hurt model performance.
>
> | Dataset | MIMIC AUROC |
> |-|-|
> | BNN + Isotropic | 69.05 $\pm$ 1.28 |
> | Banana | 73.63 $\pm$ 0.86 |
> | Banana (misspecified) | 71.92 $\pm$ 0.73  |
>
> We thank the reviewer again for their time and consideration! We are more than happy to address any other concerns they might have.

---

> > ### Author Response · Authors · 2024-11-30
> > **Hope to hear back soon**
> >
> > Dear Reviewer Sfqk,
> >
> > In our response above, we have tried to address all your comments and concerns. To summarize, we have:
> > * Added comparisons for uncertainty quantification
> > * Added a new experiment with a larger architecture of a ResNet on the Waterbirds dataset
> > * Added in a comparison to the Pretrain Your Loss baseline, showing that Banana is superior
> > * Added in all citations regarding missing prior work
> > * Added an experiment showing that Banana can be safely used with a misspecified prior
> > * Added a clarification on Banana’s computational efficiency in comparison to incorporating domain knowledge in the posterior
> >
> > Thank you again for taking the time to review our work and we hope to hear back from you soon. Please let us know if you have any additional questions!

---

> ### Comment · Reviewer_Sfqk · 2024-12-01
>
> Thank you for your detailed reply and additional experiments. Although your additional experiments and explanations help me to increase my score but I want to mention that it is really unhappy to get some responses from authors almost in the end of discussion periods. And it could hinders active discussion between the authors and reviewers. That's why I was planning to not responds to your rebuttal. But as I like your motivations and additional experiments, I will increase score to 6.

---

> > ### Author Response · Authors · 2024-12-01
> >
> > Thank you for your prompt reply and your continued engagement. We apologize for not posting our response earlier in the discussion period. We appreciate your understanding and will note this for the future.

---

### Official Review · Reviewer_GCqN · 2024-10-31

**Soundness:** 2
**Presentation:** 3
**Contribution:** 3
**Rating:** 6
**Confidence:** 4

**Summary:**

This work presents framework to learn prior that integrates domain knowledge in BNN. In detail, it expresses domain knowledge as a loss function and optimize the variational objective. It also suggests transferring learned priors between different model architectures, based on Moment Matching and Maximum Mean Discrepancy.

**Strengths:**

**[S1]** The idea of incorporating general forms of domain knowledge into prior via loss function is interesting.

**[S2]** It is novel and compelling approach transferring learned priors across different model architectures.

**[S3]** The experimental results support that the proposed methodology effectively learns prior.

**Weaknesses:**

I believe the proposed methods are solid, but there is chance to elevate the persuasive strength with experiments.



**[W1]** All experiments were conducted on a 2-layer MLP model. Performances on larger model would give more credibility to the paper. Specifically, while it may be challenging to evaluate performance on ViT, assessing this methodology’s impact on models like ResNet would further validate its potential for performance enhancement.



**[W2]** Experiments on a larger dataset seem necessary. Although increasing the data volume reduces the influence of the prior, making it difficult to evaluate the effect of the learned prior using the proposed method, the machine learning and deep learning communities widely accept the use of larger models and datasets. The paper's contribution can further be amplified by demonstrating that this methodology remains effective under larger dataset.



**[W3]** The paper lacks recent baselines. For prior-related baselines, it employs isotropic Gaussian, Gaussian optimized via Laplace, and GP prior. However, the informative prior method *Pre-train Your Loss* [1], a significant advancement in this area, should definitely be included.



**[W4]** The writing (notations) could also be improved.

- $x_{i}^{\prime}$ is not defined before Equation (1).
- I believe that $X^\prime = \{ x_1, ..., x_m\}$ in line 283-284 can be clear by rewriting as $X^\prime = \{ x_1^\prime, ..., x_m^\prime\}$



---
[1] Ravid Shwartz-Ziv, Micah Goldblum, Hossein Souri, Sanyam Kapoor, Chen Zhu, Yann LeCun, and Andrew G Wilson. Pre-Train Your Loss: Easy Bayesian Transfer Learning with Informative Priors. Advances in Neural Information Processing Systems, 2022.

**Questions:**

I appreciate for authors to conduct a good study on an intriguing topic. I have a few questions after reading this paper.

**[Q1]** What value was used for the rank $r$ in the experiments?


**[Q2]** In Figure 3, when $K = 10$, the performance is the best, then declines, and eventually rises again. What do you think causes this phenomenon?


**[Q3]** In Section 5.3, I understood that a multi-modal distribution was used to represent a complex distribution. However, I believe that the proposed methodology for learning a general informative prior is inherently complex (as shown in Figure 2). Therefore, I’m having difficulty understanding how this experiment supports the proposed methodology. Could you please explain this in more detail? If I misunderstood this section, I would appreciate any additional clarification.

---

> ### Author Response · Authors · 2024-11-27
> **Author Response to Reviewer GCqN (Part 1)**
>
> We thank the reviewer for their efforts in providing a detailed review. We appreciate that you found our method "interesting" and (specifically our prior transferring as) “novel”. We hope that our new experimental results address your comments and concerns below.
>
>
> > **[W1] All experiments were conducted on a 2-layer MLP model. Performances on larger model would give more credibility to the paper. Specifically, while it may be challenging to evaluate performance on ViT, assessing this methodology’s impact on models like ResNet would further validate its potential for performance enhancement.**
>
> In response to your request, we have added an additional experiment using a larger model and a larger dataset. For the dataset, we focus on Waterbirds [3], which consists of bird images from the CUB dataset combined with backgrounds from the Places dataset. The task is a binary classification problem: determining whether an image depicts a waterbird or a landbird. However, there are spurious correlations in the training data, where landbirds predominantly appear against land backgrounds, and waterbirds against water backgrounds.
>
> The goal is to ensure that the model does not rely on background information for making predictions. To evaluate this, we measure the accuracy for each subgroup of background and label and want to maximize the worst-group accuracy. Given the known spurious correlation between image backgrounds and labels, we leverage domain knowledge, similar to the approach used in DecoyMNIST, by penalizing the gradient of the background in the images.
>
> For the model, we use a ResNet-18 architecture. Following [2], which argues that partially stochastic networks can match or even outperform fully stochastic networks, we freeze the backbone of the ResNet-18 and only train the linear head.
>
>
> | Metric  | BNN + Isotropic | Banana |
> |-|-|-|
> | **Test Accuracy** | $56.71 \pm 0.32$ |  **70.34 $\pm$ 0.26**  |
> | **Worst-Group Accuracy** |  $19.03\pm 3.96$ | **42.15 $\pm$ 1.09** |
> | **$\phi$**  |  $0.263 \pm 0.017$ | **0.034 $\pm$ 0.001** |
> | **Worst-Group $\phi$**   | $0.330 \pm 0.022$ | **0.037 $\pm$ 0.001** |
>
>
>
> Our results show that Banana achieves better accuracy and worst-group accuracy compared to a standard BNN with an isotropic Gaussian prior. Additionally, Banana exhibits lower input gradient magnitudes (lower values of $\phi$) across all data and specifically on the worst-group as well. This demonstrates the effectiveness of our approach and the utility of incorporating domain knowledge through a prior at a larger model and dataset scale.
>
>
>
> > **[W2] Experiments on a larger dataset seem necessary. Although increasing the data volume reduces the influence of the prior, making it difficult to evaluate the effect of the learned prior using the proposed method, the machine learning and deep learning communities widely accept the use of larger models and datasets. The paper's contribution can further be amplified by demonstrating that this methodology remains effective under larger dataset.**
>
> In response to your request, we provide a new comparison of Banana with the alternatives on the same datasets, except where we use the full training data.
>
> | **Method** | **DecoyMNIST Accuracy (↑)** | **DecoyMNIST φ_background** | **MIMIC-IV AUROC (↑)** | **MIMIC-IV φ_clinical** | **Pendulum L1 Loss (↓)** | **Pendulum φ_energy_damping** |
> |-|-|-|-|-|-|-|
> | BNN + Isotropic | 76.41 ± 0.71 | 1.06 ± 0.06 | 0.6981 ± 0.0003 | 0.1624 ± 0.0005 | **0.0036 ± 0.0001** | 0.0319 ± 0.0026 |
> | BNN + Laplace | 76.47 ± 0.70 | 1.14 ± 0.04 | 0.6980 ± 0.0002 | 0.1625 ± 0.0009 | 0.0043 ± 0.0006 | 0.0367 ± 0.0100 |
> | BNN + GP Prior | 75.49 ± 0.70 | 1.54 ± 0.04 | 0.6979 ± 0.0002 | 0.1628 ± 0.0008 | - | - |
> | **Banana** | **78.21 ± 0.40** | **0.44 ± 0.01** | **0.6983 ± 0.0001** | **0.1619 ± 0.0005** | 0.0041 ± 0.0007 | **0.0025 ± 0.0010** |
>
> The sizes of the training data for each dataset is:
>
> | **Dataset** | **DecoyMNIST** | **MIMIC-IV** | **Pendulum** |
> |-|-|-|-|
> | | 30000 |10915 | 9000 |
>
> As expected, we observe less impact of using an informative prior with a larger number of labeled data. However, in this case, Banana still outperforms the alternatives across all tasks.

---

> ### Author Response · Authors · 2024-11-27
> **Author Response to Reviewer GCqN (Part 2)**
>
> > **[W3] The paper lacks recent baselines. For prior-related baselines, it employs isotropic Gaussian, Gaussian optimized via Laplace, and GP prior. However, the informative prior method Pre-train Your Loss [1]... should definitely be included.**
>
> We have added a comparison to [1], which also learns an informative prior during pretraining. We observe the following results:
>
> | Dataset | MNIST Accuracy | MNIST $\phi$ | MIMIC-IV AUROC | MIMIC-IV $\phi$ | Pendulum L1 Loss | Pendulum $\phi$ |
> |-|-|-|-|-|-|-|
> | Pretrain Loss [1] | 71.79 $\pm$ 1.35 | **1.62 $\pm$ 0.02** | 0.6670 $\pm$ 0.0101 | 0.2012 $\pm$ 0.0074 | 0.330 $\pm$ 0.003 | **0.0 $\pm$ 0.0**  |
> | Banana | **73.63 $\pm$ 0.86** | 1.65 $\pm$ 0.05 | **0.6778 $\pm$ 0.0026** | **0.1924 $\pm$ 0.0047** | **0.010 $\pm$ 0.001** | **0.0 $\pm$ 0.0** |
>
> We also provide a comparison with this task on the fairness dataset. We provide the accuracy and fairness of the posterior mean ($\pm$ 1 standard deviation). Overall, we find that the posterior mean learned via Banana is much more accurate and fair than the Pretrain Your Loss baseline, due to the likely degenerate solution learned via the original trained model on $\phi$.
>
> | | Accuracy | Group Fairness |
> |-|-|-|
> | Pretrain Your Loss [1] | 74.49 $\pm$ 0.63 | 0.0175 $\pm$ 0.003 |
> | Banana | **76.68 $\pm$ 0.34** | **0.00012 $\pm$ 0.00016** |
>
> Since we cannot provide figures via OpenReview, we present the figure of the whole distributions in our revision in Appendix D.
>
> [1] Shwartz-Ziv, et. al. Pre-Train Your Loss: Easy Bayesian Transfer Learning with Informative Priors
>
> > **[W4] The writing (notations) could also be improved. Xi′ is not defined before Equation (1). I believe that X′=x1,...,xm in line 283-284 can be clear by rewriting as X′=x1′,...,xm′**
>
> Thank you for this suggestion. We have made this change in our revision.
>
> > **[Q1] What value was used for the rank in the experiments?**
>
> We used a rank of 30 in our experiments. We would also like to highlight that we already include an ablation in our paper in Appendix A.3 that studies the impact of using a different value for the hyperparameter of rank in our prior approximation.
>
> > **[Q2] In Figure 3, when K=10, the performance is the best, then declines, and eventually rises again. What do you think causes this phenomenon?**
>
> We think that after K = 10, the prior approximation is overparameterized and can overfit to some information in the prior’s training data, which causes the decrease in performance. After K >= 15, we think this behavior is likely due to randomness in the experiment (e.g., initialization, training data order) as any of these slight decline and then slight rise in performance are contained within the error bars.
>
>
> > **[Q3] In Section 5.3, I understood that a multi-modal distribution was used to represent a complex distribution. However, I believe that the proposed methodology for learning a general informative prior is inherently complex... Therefore, I’m having difficulty understanding how this experiment supports the proposed methodology. Could you please explain this in more detail?.**
>
> This experiment further studies by increasing the complexity of our general informative prior. Rather than using a **unimodal** low-rank Multivariate Gaussian distribution, here we experiment with a **mixture** of low-rank Multivariate Gaussians, which is multi-modal. Therefore, this ablation studies the role of increasing complexity on how well using our informative prior helps increase performance.
>
> Thank you again for your efforts in providing a detailed review. We hope you will consider raising your score if your concerns have been addressed.

---

> > ### Comment · Reviewer_GCqN · 2024-11-28
> >
> > I appreciate for your kind reply.
> > My concerns and questions have been addressed, and i've decided to raise your score by one point.
> > I wish you the best of luck.

---

### Official Review · Reviewer_F9oY · 2024-11-03

**Soundness:** 3
**Presentation:** 3
**Contribution:** 2
**Rating:** 5
**Confidence:** 4

**Summary:**

This paper proposes a framework for integrating domain knowledge into a prior over neural network weights. It uses variational inference to learn a low-rank Gaussian prior in weight space, guided by a domain knowledge loss that depends only on input data. The paper proposes four practical losses that incorporate different types of domain knowledge. It also proposes methods for transferring priors between different architectures.

**Strengths:**

- The paper is well-motivated, and the approach is clean. Incorporating domain knowledge into informative priors is an interesting direction for Bayesian deep learning.
- The paper proposes four general domain knowledge losses that supplement the available training data in meaningful ways.
- The paper demonstrates that the learned prior is better aligned with domain knowledge and often improves predictive performance compared to standard priors.

**Weaknesses:**

- My main concern is that I'm not convinced that the best way to use the proposed domain knowledge losses is to learn a prior over weights upfront. As the paper states in page 4, we could also use these losses to regularize the training process. You could similarly train a BNN with a likelihood function that incorporates these losses. Could the authors explain why learning a prior is better than these alternatives? Section A.5 empirically compares this alternative somewhat. Still, I think a more thorough comparison would be to do a wide sweep over values of the regularization coefficient and report (performance, domain loss) for each.
- The paper mentions uncertainty as a primary motivation for using BNNs (abstract, intro, ...), but the evaluation does not measure uncertainty; this is important since the prior learned by Banana may improve performance over standard priors at the cost of worse uncertainty estimation.
- Fairness loss involves subgroup information, which is known to significantly improve fairness metrics when used during training [1]. For the experiment on Folktables, I think the comparison to the baseline is not entirely fair because the baseline doesn't have access to the subgroup information.

[1] Sagawa, Shiori, et al. "Distributionally robust neural networks for group shifts: On the importance of regularization for worst-case generalization." arXiv preprint arXiv:1911.08731 (2019).

Minor comment:
- (just out of curiosity) Is Banana an acronym for something?

**Questions:**

Please see weaknesses section above.

---

> ### Author Response · Authors · 2024-11-27
> **Author Reponse to Reviewer F9oY**
>
> We thank the reviewer for their efforts in providing a detailed review. We appreciate that you found our method "well-motivated" and in an interesting direction of research. We address your concerns below.
>
> > **My main concern is that I'm not convinced that the best way to use the proposed domain knowledge losses is to learn a prior over weights upfront… Could the authors explain why learning a prior is better than these alternatives?**
>
> One key benefit of learning a prior that incorporates domain knowledge, rather than incorporating it during posterior samples is computation efficiency. We only need to learn the prior a single time, which involves training on $\phi$ once. However, when using domain knowledge at the stage of sampling from the posterior, each sample must be trained with $\phi$. Since BNNs benefit from averaging multiple posterior samples (Figure A2 in the Appendix), one needs to incorporate $\phi$ everytime we draw a new sample from the posterior distribution. This is $k$ times more expensive when $k$ is the number of posterior samples. Thus, computational efficiency is actually a benefit of our approach.
>
> Another reason for our focus on learning this information in priors is the open-source release of informative priors, which others could take and incorporate later in their own downstream applications. This could provide general benefits to their model, without having to use multiple forms of domain knowledge at the posterior stage.
>
> Finally, we remark that learning an informative prior is an ongoing challenge in the Bayesian community. Our work introduces a novel approach to incorporating broad domain knowledge into the prior of a Bayesian neural network (BNN), such as physics-based rules or background exclusion in images—concepts that have not been extensively explored in prior work. This contribution serves as a foundational step toward integrating richer, more practical domain knowledge into priors, which we believe opens up exciting opportunities for future research.
>
> > **The paper mentions uncertainty as a primary motivation for using BNNs (abstract, intro, ...), but the evaluation does not measure uncertainty**
>
>
> In response to your request, we provide an additional experiment comparing the ECE of Banana to the BNN + Isotropic Gaussian baseline.
>
> | Method | MNIST | MIMIC | ACS |
> |-|-|-|-|
> | BNN + Isotropic | 0.24 $\pm$ 0.01 | **0.27 $\pm$ 0.01** | 0.0053 $\pm$ 0.0007 |
> | Banana (ours) | **0.19 $\pm$ 0.00** | **0.27 $\pm$ 0.01** | **0.0049 $\pm$ 0.0009** |
>
>
> We observe that Banana achieves slightly better or comparable calibration performance in terms of ECE on all tasks. This shows that the model has successfully incorporated our desirable domain knowledge in a Bayesian setting.
>
>
> > **For the experiment on Folktables, I think the comparison to the baseline is not entirely fair because the baseline doesn't have access to the subgroup information.**
>
> The main purpose of this experiment is to demonstrate that our approach can indeed successfully incorporate domain knowledge into a BNN via a learned prior. While the baseline does not use the subgroup information, we show that our method can incorporate this information, leading to a more fair and better-performing posterior distribution. In our new experiment with the Pretrain Your Loss baseline, we show that we still outperform this approach, which indeed does use subgroup information (via the same $\phi$). The posterior means for this comparison are provided below, and we present the full distributions in Appendix C.
>
> | | Accuracy | Group Fairness |
> |-|-|-|
> | Pretrain Your Loss [1] | 74.49 $\pm$ 0.63 | 0.0175 $\pm$ 0.003 |
> | Banana | **76.68 $\pm$ 0.34** | **0.00012 $\pm$ 0.00016** |
>
> > **(just out of curiosity) Is Banana an acronym for something?**
>
> Banana is an acronym taken from **Ba**yesian **N**eural **N**etworks with Dom**a**in Knowledge.
>
> We thank the reviewer again for their time and consideration! We are more than happy to address any other concerns they might have.

---

> > ### Author Response · Authors · 2024-11-30
> > **Hope to hear back soon**
> >
> > Dear Reviewer F9oY,
> >
> > In our response above, we have tried to address all your comments and concerns. To summarize, we have:
> > * Added comparisons for uncertainty quantification
> > * Added in a comparison on the Folktables dataset of the Pretrain Your Loss baseline, which does use subgroup information
> > * Added clarifications on the benefits of learning domain knowledge via a prior rather than incorporating it via the posterior
> >
> > Thank you again for taking the time to review our work and we hope to hear back from you soon. Please let us know if you have any additional questions!

---

### Official Review · Reviewer_UQ77 · 2024-11-04

**Soundness:** 2
**Presentation:** 2
**Contribution:** 3
**Rating:** 5
**Confidence:** 4

**Summary:**

This paper proposes a framework for incorporating general form of domain knowledge into Bayesian neural network (BNN) through learned informative priors. These domain knowledge priors are specified via loss functions that measures the alignment of a particular model to the desired domain knowledge, and learned through variational inference. Empirical evaluation is performed using Stochastic Gradient Langevin Dynamics (SGLD) as the baseline with experiments on a 2-layer feedforward neural network, demonstrating the model with domain knowledge priors achieves better predictive performance than BNNs with commonly used priors (such as isotropic, Gaussian).

**Strengths:**

* The paper is well-motivated to specify informed priors in BNNs that reflects the relevant domain knowledge and mitigate undesirable biases.

* The proposed framework enables the integration of generic forms of domain knowledge such as physics rules, fairness, healthcare knowledge into BNN prior, and also proposes strategy to transfer the priors to other models without the need to relearn a new prior every time.

**Weaknesses:**

* One of the main objective of Bayesian neural network (BNN) is uncertainty quantification (UQ) in their predictions. However, this paper does not address the ability of BNNs to reliably quantify model uncertainty in the study and experimental evaluation. The experiments conducted in the study are limited to evaluating predictive accuracy and domain knowledge surrogate loss, neglecting the critical UQ aspect of BNNs. This raises concerns about the evaluation of BNNs with the proposed informed prior, and the reliability of the model's uncertainty estimates, which are essential. I encourage the authors to include a study evaluating the quality of model uncertainty estimates using commonly used metrics such as Expected Calibration Error, AUROC for out-of-distribution detection etc.

* The paper claims to propose a strategy for transferring learned informative priors across different neural network architectures. However, the experiments conducted to validate this claim are limited to transferring priors between two 2-layer feed forward neural networks with different hidden dimension sizes. This raises concerns about the generalizability of the proposed strategy to significantly different architectures, such as Convolutional Neural Networks (CNNs) or other architectures used in the Bayesian deep learning literature. I encourage the authors to evaluate transferring learned priors between MLPs and CNNs, or vice-versa.

* The lack of experimentation with a broader range of neural network architectures beyond a small 2-layer feedforward model leaves open questions about the effectiveness of the proposed method to other model architectures and it's scalability to larger models. Also, the motivation for the choice of datasets used in the empirical experiments is not clear. The datasets selected are uncommon in the Bayesian deep learning literature, which makes it difficult to compare the results with existing studies and to assess the relevance and robustness of the proposed method.

**Questions:**

* How are the variational parameters of the BNN initialized? Does initialization of the variational posterior q(w) play a role in addressing the domain knowledge awareness, or specifying the prior p(w) is sufficient?

* The results of Folktables dataset are presented in the transferrinf priors experiments in Table 2. Any reason for not providing the results of this dataset in Table 1?

* Can the authors clarify their motivation for choosing the specific datasets used in the experiments?

---

> ### Author Response · Authors · 2024-11-27
> **Author Response to Reviewer UQ77**
>
> We thank the reviewer for their efforts in providing a detailed review. We address your concerns below.
>
> > **I encourage the authors to include a study evaluating the quality of model uncertainty estimates...**
>
> In response to your request, we provide an additional experiment comparing the ECE of Banana to the BNN + Isotropic Gaussian baseline.
>
> | Method | MNIST | MIMIC | ACS |
> |-|-|-|-|
> | BNN + Isotropic | 0.24 $\pm$ 0.01 | **0.27 $\pm$ 0.01** | 0.0053 $\pm$ 0.0007 |
> | Banana (ours) | **0.19 $\pm$ 0.00** | **0.27 $\pm$ 0.01** | **0.0049 $\pm$ 0.0009** |
>
>
> We observe that Banana achieves slightly better or comparable calibration performance in terms of ECE on all tasks. This shows that the model has successfully incorporated our desirable domain knowledge in a Bayesian setting.
>
> > **The lack of experimentation with a broader range of neural network architectures beyond a small 2-layer feedforward model leaves open questions about the effectiveness... and it's scalability to larger models.**
>
> In response to your request, we have added an additional experiment using a larger model and a larger dataset. For the dataset, we focus on Waterbirds [3], which consists of bird images from the CUB dataset combined with backgrounds from the Places dataset. The task is a binary classification problem: determining whether an image depicts a waterbird or a landbird. However, there are spurious correlations in the training data, where landbirds predominantly appear against land backgrounds, and waterbirds against water backgrounds.
>
> The goal is to ensure that the model does not rely on background information for making predictions. To evaluate this, we measure the accuracy for each subgroup of background and label and want to maximize the worst-group accuracy. Given the known spurious correlation between image backgrounds and labels, we leverage domain knowledge, similar to the approach used in DecoyMNIST, by penalizing the gradient of the background in the images.
>
> For the model, we use a ResNet-18 architecture. Following [2], which argues that partially stochastic networks can match or even outperform fully stochastic networks, we freeze the backbone of the ResNet-18 and only train the linear head.
>
> | Metric  | BNN + Isotropic | Banana |
> |-|-|-|
> | **Test Accuracy** | $56.71 \pm 0.32$ |  **70.34 $\pm$ 0.26**  |
> | **Worst-Group Accuracy** |  $19.03\pm 3.96$ | **42.15 $\pm$ 1.09** |
> | **$\phi$**  |  $0.263 \pm 0.017$ | **0.034 $\pm$ 0.001** |
> | **Worst-Group $\phi$**   | $0.330 \pm 0.022$ | **0.037 $\pm$ 0.001** |
>
> Our results show that Banana achieves better accuracy and worst-group accuracy compared to a standard BNN with an isotropic Gaussian prior. Additionally, Banana exhibits lower input gradient magnitudes (lower values of $\phi$) across all data and specifically on the worst-group as well. This demonstrates the effectiveness of our approach and the utility of incorporating domain knowledge through a prior at a larger model and dataset scale.
>
>
> > **motivation for the choice of datasets used in the empirical experiments is not clear… Can the authors clarify their motivation for choosing the specific datasets... ?****
>
> A key distinction between our work and prior works on BNNs is that we are trying to incorporate useful domain knowledge or side information into our BNNs. Therefore, we examine a variety of different settings where such domain knowledge is available – and these settings are exactly the datasets that we consider in our paper. A requirement for our method is the ability to represent useful domain knowledge as a loss function $\phi$, and we provide such losses on each dataset in Section 5.1.
>
> > **How are the variational parameters of the BNN initialized? Does initialization of the variational posterior q(w) play a role in addressing the domain knowledge awareness, or specifying the prior p(w) is sufficient?**
>
> The variational parameters of the BNN are initialized from a random normal distribution. The initialization does not play a significant role and specifying the prior $p(w)$ is what defines the domain knowledge awareness.
>
> > **The results of Folktables dataset are presented in the transferring priors experiments in Table 2. Any reason for not providing the results of this dataset in Table 1?**
>
> We did not include the Folktables results in Table 1, since this experiment and dataset looks to demonstrate a different point. For the datasets in Table 1, the domain knowledge aims to help the accuracy of our BNNs predictions. However, for Folktables, the domain knowledge we are trying to incorporate is fairness, which has been shown to be at odds with accuracy/performance [1]. Thus, we rather display the results on Folktables as a plot to illustrate both the performance and fairness of the method.
>
> We thank the reviewer again for their time and consideration! We are more than happy to address any other concerns they might have.

---

> > ### Author Response · Authors · 2024-11-30
> > **Hope to hear back soon**
> >
> > Dear Reviewer UQ77,
> >
> > In our response above, we have tried to address all your comments and concerns. To summarize, we have:
> > * Added comparisons for uncertainty quantification
> > * Added a new experiment with a larger architecture of a ResNet on the Waterbirds dataset
> > * Added clarifications on the choice of datasets due to the presence of domain knowledge
> > * Added a clarification on the Folktables results and figure
> >
> > Thank you again for taking the time to review our work and we hope to hear back from you soon. Please let us know if you have any additional questions!

---

> > > ### Comment · Reviewer_UQ77 · 2024-12-01
> > >
> > > I thank the authors for the responses. I have no additional questions.

---

> > > > ### Author Response · Authors · 2024-12-03
> > > >
> > > > Thank you! If we have resolved your questions and concerns, we hope you will consider raising your initial evaluation.

---

### Official Review · Reviewer_38LC · 2024-11-10

**Soundness:** 2
**Presentation:** 3
**Contribution:** 2
**Rating:** 5
**Confidence:** 4

**Summary:**

This paper proposes a approach for incorporating domain knowledge into priors for Bayesian neural networks (BNNs).
The domain knowledge is represented as a loss function phi that measures how well a model aligns with the given knowledge. Using variational inference phi is used to reweight the prior distribution over neural network weights.

**Strengths:**

The manuscript tries to address the important topic of informative priors for efficiently modeling Bayesian inference to incorporate domain knowledge.

**Weaknesses:**

- The theoretical justification for phi loss incorporating domain knowledge is not clear; this proposed formulation resembles an empirical Bayes setup where informative priors are learned from the data.
- There is a large body of work on informative priors that considers techniques such as empirical Bayes and hierarchical Bayes to incorporate domain knowledge.
- In the results section, it is unclear why the comparison of phi values against the selected datasets indicates the incorporation of domain knowledge.
- It is unclear how to interpret whether the model has successfully incorporated domain knowledge in a Bayesian setting without comparing standard uncertainty quantification metrics.
- While the method claims low complexity, it involves learning an informative prior through variational inference, which may add computational overhead compared to using standard uninformative priors. The paper does not discuss the computational costs, aside from a brief mention in the Appendix.
- The selected problems involve very small datasets (<50 samples) and toy problems, but practical examples with approximately 1,000 samples are needed to demonstrate effectiveness.

**Questions:**

Please see the above section.

---

> ### Author Response · Authors · 2024-11-27
> **Author Response to Reviewer 38LC**
>
> Thank you for your time and effort in providing a detailed review! We address your concerns below.
>
> > **The theoretical justification for phi loss incorporating domain knowledge is not clear; this proposed formulation resembles an empirical Bayes setup where informative priors are learned from the data.**
>
> In our setting, we represent our domain knowledge as a loss function $\phi$, which captures how well a model reflects this domain knowledge. Concrete examples of this are provided in Section 5.1. While indeed there may be some similarities between standard empirical Bayes approaches, where priors are learned from the data, the *key distinction* in our setting is exactly this loss value $\phi$. Incorporating domain knowledge in this sort of fashion, to the best of our knowledge, has not been tackled by existing empirical Bayes approaches or by other works in the literature.
>
>
> > **There is a large body of work on informative priors that considers techniques such as empirical Bayes and hierarchical Bayes to incorporate domain knowledge.**
>
> We believe that while there is some work that learns informed priors (from the observed data), prior works have not tried to incorporate additional side information / domain knowledge such as fairness, ignoring spurious features, or rules from physics into BNNs.
>
> > **It is unclear how to interpret whether the model has successfully incorporated domain knowledge in a Bayesian setting without comparing standard uncertainty quantification metrics.**
>
> In response to your request, we provide an additional experiment comparing the ECE of Banana to the BNN + Isotropic Gaussian baseline.
>
> | Method | MNIST | MIMIC | ACS |
> |-|-|-|-|
> | BNN + Isotropic | 0.24 $\pm$ 0.01 | **0.27 $\pm$ 0.01** | 0.0053 $\pm$ 0.0007 |
> | Banana (ours) | **0.19 $\pm$ 0.00** | **0.27 $\pm$ 0.01** | **0.0049 $\pm$ 0.0009** |
>
> We observe that Banana achieves slightly better or comparable calibration performance in terms of ECE on all tasks. This shows that the model has successfully incorporated our desirable domain knowledge in a Bayesian setting.
>
>
> > **While the method claims low complexity, it involves learning an informative prior through variational inference, which may add computational overhead compared to using standard uninformative priors. The paper does not discuss the computational costs, aside from a brief mention in the Appendix.**
>
> While our approach does have an added computation overhead of learning the prior when compared to standard uninformative priors, this is only a small single-time cost of training a prior. Indeed, this also has many benefits of introducing desirable properties into our BNN, while standard priors do not encode any of this useful information.
>
> > **The selected problems involve very small datasets (<50 samples) and toy problems, but practical examples with approximately 1,000 samples are needed to demonstrate effectiveness.**
>
> In response to your request, we provide a new comparison of Banana with the alternatives on the same datasets, except where we use the full training data.
>
> | **Method** | **DecoyMNIST Accuracy (↑)** | **DecoyMNIST φ_background** | **MIMIC-IV AUROC (↑)** | **MIMIC-IV φ_clinical** | **Pendulum L1 Loss (↓)** | **Pendulum φ_energy_damping** |
> |-|-|-|-|-|-|-|
> | BNN + Isotropic | 76.41 ± 0.71 | 1.06 ± 0.06 | 0.6981 ± 0.0003 | 0.1624 ± 0.0005 | **0.0036 ± 0.0001** | 0.0319 ± 0.0026 |
> | BNN + Laplace | 76.47 ± 0.70 | 1.14 ± 0.04 | 0.6980 ± 0.0002 | 0.1625 ± 0.0009 | 0.0043 ± 0.0006 | 0.0367 ± 0.0100 |
> | BNN + GP Prior | 75.49 ± 0.70 | 1.54 ± 0.04 | 0.6979 ± 0.0002 | 0.1628 ± 0.0008 | - | - |
> | **Banana** | **78.21 ± 0.40** | **0.44 ± 0.01** | **0.6983 ± 0.0001** | **0.1619 ± 0.0005** | 0.0041 ± 0.0007 | **0.0025 ± 0.0010** |
>
>
> The sizes of the training data for each dataset is:
>
> | | **DecoyMNIST** | **MIMIC-IV** | **Pendulum** |
> |-|-|-|-|
> |Number of Training Data | 30000 |10915 | 9000 |
>
> As expected, we observe less impact of using an informative prior with a larger number of labeled data. However, in this case, Banana still outperforms the alternatives across all tasks.
>
> We thank the reviewer again for their time and consideration! We are more than happy to address any other concerns they might have.

---

> > ### Author Response · Authors · 2024-11-30
> > **Hope to hear back soon**
> >
> > Dear Reviewer 38LC,
> >
> > In our response above, we have tried to address all your comments and concerns. To summarize, we have:
> > * Added comparisons for uncertainty quantification
> > * Added a new experiment with a larger architecture of a ResNet on the Waterbirds dataset
> > * Added results on larger labeled dataset sizes
> > * Clarified the differences between Banana and empirical Bayes
> > * Added a clarification on the slight increase in Banana’s computational overhead
> >
> >
> > Thank you again for taking the time to review our work and we hope to hear back from you soon. Please let us know if you have any additional questions!

---

> > > ### Comment · Reviewer_38LC · 2024-12-03
> > >
> > > I thank the authors for responding the comments. After reviewing all the feedback/comments, I'll keep my current rating.

---

### Author Response · Authors · 2024-11-27
**Overall Author Response**

We thank the reviewers for their efforts in providing detailed reviews. We appreciate that the reviewers found our paper “well-motivated” [UQ77, F9oY] in tackling an “important” [38LC] problem setting. We also are glad that the reviewers find our prior transferring approach as “novel and compelling” [GCqN].

We have added 5 new experiments (see Section 5.4 and Appendices A-D) and address your comments and concerns below.

---

### Meta-Review · Area_Chair_bzp3 · 2024-12-20

**Metareview:**

The paper presents a novel approach to incorporating domain knowledge into the prior distribution of Bayesian Neural Networks (BNNs) through variational inference, which can be used to encode various inductive biases, such as fairness and physics-based rules. While the reviewers found the paper to be well-written and the approach to be interesting and novel, they raised several concerns, including the lack of experiments on larger models and datasets, the need for more comparisons to recent baselines, and the potential limitations of the loss-based formulation. However, the authors effectively addressed these concerns by providing additional experiments, including comparisons to recent baselines, such as Pretrain Your Loss, and experiments on larger models, including a ResNet-18 architecture on the Waterbirds dataset, which demonstrated the effectiveness of their approach in incorporating domain knowledge and improving model performance, particularly in terms of fairness and worst-group accuracy. Overall, the authors' responses and additional experiments have clearly strengthened the paper. However, after discussion with the authors and among themselves, the reviewers find the paper to still be very borderline, with two reviewers leaning towards acceptance and three towards rejection. We will therefore need to reject the paper in its current form. We would still like to encourage the authors to resubmit an improved version of the paper in the future.

**Additional Comments On Reviewer Discussion:**

see above

---

### Decision · Program_Chairs · 2025-01-22

Reject